# A THEORY OF REPRESENTATION LEARNING IN NEURAL NETWORKS GIVES A DEEP GENERALISATION OF KERNEL METHODS

## ABSTRACT

The successes of modern deep machine learning methods are founded on their ability to transform inputs across multiple layers to build good high-level representations. It is therefore critical to understand this process of representation learning. However, standard theoretical approaches (formally NNGPs) involving infinite width limits eliminate representation learning. We therefore develop a new infinite width limit, the Bayesian representation learning limit, that exhibits representation learning mirroring that in finite-width models, yet at the same time, retains some of the simplicity of standard infinite-width limits. In particular, we show that Deep Gaussian processes (DGPs) in the Bayesian representation learning limit have exactly multivariate Gaussian posteriors, and the posterior covariances can be obtained by optimizing an interpretable objective combining a log-likelihood to improve performance with a series of KL-divergences which keep the posteriors close to the prior. We confirm these results experimentally in wide but finite DGPs. Next, we introduce the possibility of using this limit and objective as a flexible, deep generalisation of kernel methods, that we call deep kernel machines (DKMs). Like most naive kernel methods, DKMs scale cubically in the number of datapoints. We therefore use methods from the Gaussian process inducing point literature to develop a sparse DKM that scales linearly in the number of datapoints. Finally, we extend these approaches to NNs (which have non-Gaussian posteriors) in the Appendices.

## 1 INTRODUCTION

The successes of modern machine learning methods from neural networks (NNs) to deep Gaussian processes (DGPs Damianou & Lawrence, 2013; Salimbeni & Deisenroth, 2017) is based on their ability to use depth to transform the input into high-level representations that are good for solving difficult tasks (Bengio et al., 2013; LeCun et al., 2015). However, theoretical approaches using infinite limits to understand deep models struggle to capture representation learning. In particular, there are two broad families of infinite limit, and while they both use kernel-matrix-like objects they are ultimately very different. The neural network Gaussian process (NNGP Neal, 1996; Lee et al., 2017; Matthews et al., 2018) applies to Bayesian models like Bayesian neural networks (BNNs) and DGPs and describes the representations at each layer (formally, the NNGP kernel is raw second moment of the activities). In contrast, the neural tangent kernel (NTK Jacot et al., 2018) is a very different quantity that involves gradients, and describes how predictions at all datapoints change if we do a gradient update on a single datapoint. As such, the NNGP and NTK are suited to asking very different theoretical questions. For instance, the NNGP is better suited to understanding the transformation of representations across layers, while the NTK is better suited to understanding how predictions change through NN training.

While challenges surrounding representation learning have recently been addressed in the NTK setting Yang & Hu (2020), we are the first to address this challenge in the NNGP setting.

At the same time, kernel methods (Smola & Schölkopf, 1998; Shawe-Taylor & Cristianini, 2004; Hofmann et al., 2008) were a leading machine learning approach prior to the deep learning revolution Krizhevsky et al. (2012). However, kernel methods were eclipsed by *deep* NNs because depth

gives NNs the flexibility to learn a good top-layer representation (Aitchison, 2020). In contrast, in a standard kernel method, the kernel (or equivalently the representation) is highly inflexible — there are usually a few tunable hyperparameters, but nothing that approaches the enormous flexibility of the top-layer representation in a deep model. There is therefore a need to develop flexible, deep generalisations of kernel method. Remarkably, our advances in understanding representation learning in DGPs give such a flexible, deep kernel method.

## 2 CONTRIBUTIONS

- We present a new infinite width limit, the Bayesian representation learning limit, that retains representation learning in deep Bayesian models including DGPs. The key insight is that as the width goes to infinity, the prior becomes stronger, and eventually overwhelms the likelihood. We can fix this by rescaling the likelihood to match the prior. This rescaling can be understood in a Bayesian context as copying the labels (Sec. 4.3).
- We show that in the Bayesian representation learning limit, DGP posteriors are exactly zero-mean multivariate Gaussian, $\mathrm{P}\left(\mathbf{f}_\lambda^\ell | \mathbf{X}, \mathbf{y}\right) = \mathcal{N}\left(\mathbf{f}_\lambda^\ell; \mathbf{0}, \mathbf{G}_\ell\right)$ where $\mathbf{f}_\lambda^\ell$, is the activation of the $\lambda$th feature in layer $\ell$ for all inputs (Sec. 4.4 and Appendix D).
- We show that the posterior covariances can be obtained by optimizing the "deep kernel machine objective",

$$\mathcal{L}(\mathbf{G}_1, \ldots, \mathbf{G}_L) = \log \mathrm{P}\left(\mathbf{Y} | \mathbf{G}_L\right) - \sum_{\ell=1}^L \nu_\ell \, \mathrm{D}_{\mathrm{KL}}\left(\mathcal{N}\left(\mathbf{0}, \mathbf{G}_\ell\right) \| \mathcal{N}\left(\mathbf{0}, \mathbf{K}(\mathbf{G}_{\ell-1})\right)\right),$$

where $\mathbf{G}_\ell$ are the posterior covariances, $\mathbf{K}(\mathbf{G}_{\ell-1})$ are the kernel matrices, and $\nu_\ell$ accounts for any differences in layer width (Sec. 4.3).
- We give an interpretation of this objective, with $\log \mathrm{P}\left(\mathbf{Y} | \mathbf{G}_L\right)$ encouraging improved performance, while the KL-divergence terms act as a regulariser, keeping posteriors, $\mathcal{N}\left(\mathbf{0}, \mathbf{G}_\ell\right)$, close to the prior, $\mathcal{N}\left(\mathbf{0}, \mathbf{K}(\mathbf{G}_{\ell-1})\right)$ (Sec. 4.5).
- We introduce a sparse DKM, which takes inspiration GP inducing point literature to obtain a practical, scalable method that is linear in the number of datapoints. In contrast, naively computing/optimizing the DKM objective is cubic in the number of datapoints (as with most other naive kernel methods; Sec. 4.7).
- We extend these results to BNNs (which have non-Gaussian posteriors) in Appendix A.

## 3 RELATED WORK

Our work is focused on DGPs and gives new results such as the extremely simple multivariate Gaussian form for DGP true posteriors. As such, our work is very different from previous work on NNs, where such results are not available. There are at least three families of such work. First, there is recent work on representation learning in the very different NTK setting (Jacot et al., 2018; Yang, 2019; Yang & Hu, 2020) (see Sec. 1). In contrast, here we focus on NNGPs (Neal, 1996; Williams, 1996; Lee et al., 2017; Matthews et al., 2018; Novak et al., 2018; Garriga-Alonso et al., 2018; Jacot et al., 2018), where the challenge of representation learning has yet to be addressed. Second, there is a body of work using methods from physics to understand representation learning in neural networks (Antognini, 2019; Dyer & Gur-Ari, 2019; Hanin & Nica, 2019; Aitchison, 2020; Li & Sompolinsky, 2020; Yaida, 2020; Naveh et al., 2020; Zavatone-Veth et al., 2021; Zavatone-Veth & Pehlevan, 2021; Roberts et al., 2021; Naveh & Ringel, 2021; Halverson et al., 2021). This work is focuses on perturbational, rather than variational methods. Third, there is a body of theoretical work including (Mei et al., 2018; Nguyen, 2019; Sirignano & Spiliopoulos, 2020a;b; Nguyen & Pham, 2020) which establishes properties such as convergence to the global optimum. This work is focused on two-layer (or one-hidden layer network) networks, and like the NTK, considers learning under SGD rather than Bayesian posteriors.

Another related line of work uses kernels to give a closed-form expression for the weights of a neural network, based on a greedy, layerwise objective (Wu et al., 2022). This work differs in that it uses the HSIC objective, and therefore does not have a link to DGPs or Bayesian neural networks, and in that it uses a greedy-layerwise objective, rather than end-to-end gradient descent.

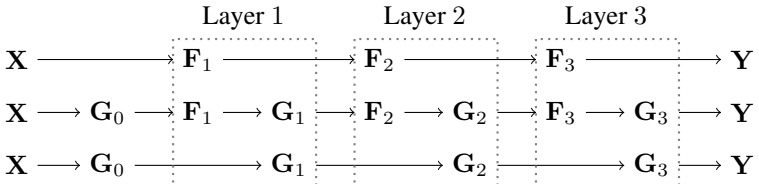

Figure 1: The graphical model structure for each of our generative models for $L = 3$. **Top**. The standard model (Eq. 1), written purely in terms of features, $\mathbf{F}_\ell$. **Middle**. The standard model, including Gram matrices as random variables (Eq. 5) **Bottom**. Integrating out the activations, $\mathbf{F}_\ell$,

## 4 RESULTS

We start by defining a DGP, which contains Bayesian NN (BNNs) as a special case (Appendix A). This model maps from inputs, $\mathbf{X} \in \mathbb{R}^{P \times \nu_0}$, to outputs, $\mathbf{Y} \in \mathbb{R}^{P \times \nu_{L+1}}$, where $P$ is the number of input points, $\nu_0$ is the number of input features, and $\nu_{L+1}$ is the number of output features. The model has $L$ intermediate layers, indexed $\ell \in \{1, \dots, L\}$, and at each intermediate layer there are $N_\ell$ features, $\mathbf{F}_\ell \in \mathbb{R}^{P \times N_\ell}$. Both $\mathbf{F}_\ell$ and $\mathbf{Y}$ can be written as a stack of vectors,

$$\mathbf{F}_\ell = (\mathbf{f}_1^\ell \quad \mathbf{f}_2^\ell \quad \cdots \quad \mathbf{f}_{N_\ell}^\ell) \qquad \mathbf{Y} = (\mathbf{y}_1 \quad \mathbf{y}_2 \quad \cdots \quad \mathbf{y}_{\nu_{L+1}}),$$

where $\mathbf{f}_\lambda^\ell \in \mathbb{R}^P$ gives the value of one feature and $\mathbf{y}_\lambda \in \mathbb{R}^P$ gives the value of one output for all $P$ input points. The features, $\mathbf{F}_1, \dots, \mathbf{F}_L$, and (for regression) the outputs, $\mathbf{Y}$, are sampled from a Gaussian process (GP) with a covariance which depends on the previous layer features (Fig. 1 top),

$$\mathrm{P}\left(\mathbf{F}_\ell | \mathbf{F}_{\ell-1}\right) = \prod_{\lambda=1}^{N_\ell} \mathcal{N}\left(\mathbf{f}_\lambda^\ell; \mathbf{0}, \mathbf{K}(\mathbf{G}(\mathbf{F}_{\ell-1}))\right) \tag{1a}$$

$$\mathrm{P}\left(\mathbf{Y} | \mathbf{F}_L\right) = \prod_{\lambda=1}^{\nu_{L+1}} \mathcal{N}\left(\mathbf{y}_\lambda; \mathbf{0}, \mathbf{K}(\mathbf{G}(\mathbf{F}_L)) + \sigma^2 \mathbf{I}\right). \tag{1b}$$

Note we only use the regression likelihood to give a concrete example; we could equally use an alternative likelihood e.g. for classification (Appendix B). The distinction between DGPs and BNNs arises through the choice of $\mathbf{K}(\cdot)$ and $\mathbf{G}(\cdot)$. For BNNs, see Appendix A. For DGPs, $\mathbf{G}(\cdot)$, which takes the features and computes the corresponding $P \times P$ Gram matrix, is

$$\mathbf{G}(\mathbf{F}_{\ell-1}) = \frac{1}{N_{\ell-1}} \sum_{\lambda=1}^{N_{\ell-1}} \mathbf{f}_\lambda^{\ell-1}(\mathbf{f}_\lambda^{\ell-1})^T = \frac{1}{N_{\ell-1}} \mathbf{F}_{\ell-1} \mathbf{F}_{\ell-1}^T. \tag{2}$$

Now, we introduce random variables representing the Gram matrices, $\mathbf{G}_{\ell-1} = \mathbf{G}(\mathbf{F}_{\ell-1})$, where $\mathbf{G}_{\ell-1}$ is a random variable representing the Gram matrix at layer $\ell - 1$, whereas $\mathbf{G}(\cdot)$ is a deterministic function that takes features and computes the corresponding Gram matrix using Eq. (2). Finally, $\mathbf{K}(\cdot)$, transforms the Gram matrices, $\mathbf{G}_{\ell-1}$ to the final kernel. Many kernels of interest are isotropic, meaning they depend only on the normalized squared distance between datapoints, $R_{ij}$,

$$K_{\text{isotropic};ij}(\mathbf{G}_{\ell-1}) = k_{\text{isotropic}}\left(R_{ij}(\mathbf{G}_{\ell-1})\right). \tag{3}$$

Importantly, we can compute this squared distance from $\mathbf{G}_{\ell-1}$, without needing $\mathbf{F}_{\ell-1}$,

$$R_{ij}(\mathbf{G}) = \frac{1}{N} \sum_{\lambda=1}^N \left(F_{i\lambda} - F_{j\lambda}\right)^2 = \frac{1}{N} \sum_{\lambda=1}^N \left(\left(F_{i\lambda}\right)^2 - 2F_{i\lambda}F_{j\lambda} + \left(F_{j\lambda}\right)^2\right)$$
$$= G_{ii} - 2G_{ij} + G_{jj}, \tag{4}$$

where $\lambda$ indexes features, $i$ and $j$ index datapoints and we have omitted the layer index for simplicity. Importantly, we are not restricted to isotropic kernels: other kernels that depend only on the Gram matrix, such as the arccos kernels from the infinite NN literature (Cho & Saul, 2009) can also be used (for further details, see Aitchison et al., 2020).

### 4.1 BNN AND DGP PRIORS CAN BE WRITTEN PURELY IN TERMS OF GRAM MATRICES

Notice that $\mathbf{F}_\ell$ depends on $\mathbf{F}_{\ell-1}$ only through $\mathbf{G}_{\ell-1} = \mathbf{G}(\mathbf{F}_{\ell-1})$, and $\mathbf{Y}$ depends on $\mathbf{F}_L$ only through $\mathbf{G}_L = \mathbf{G}(\mathbf{F}_L)$ (Eq. 1). We can therefore write the graphical model in terms of those Gram matrices (Fig. 1 middle).

$$\mathrm{P}\left(\mathbf{F}_\ell | \mathbf{G}_{\ell-1}\right) = \prod_{\lambda=1}^{N_\ell} \mathcal{N}\left(\mathbf{f}_\lambda^\ell; \mathbf{0}, \mathbf{K}(\mathbf{G}_{\ell-1})\right) \tag{5a}$$

$$\mathrm{P}\left(\mathbf{G}_\ell | \mathbf{F}_\ell\right) = \delta\left(\mathbf{G}_\ell - \mathbf{G}(\mathbf{F}_\ell)\right) \tag{5b}$$

$$\mathrm{P}\left(\mathbf{Y} | \mathbf{G}_L\right) = \prod_{\lambda=1}^{\nu_{L+1}} \mathcal{N}\left(\mathbf{y}_\lambda; \mathbf{0}, \mathbf{K}(\mathbf{G}_L) + \sigma^2 \mathbf{I}\right). \tag{5c}$$

where $\delta$ is the Dirac-delta, and $\mathbf{G}_0$ depends on $\mathbf{X}$ (e.g. $\mathbf{G}_0 = \frac{1}{\nu_0}\mathbf{X}\mathbf{X}^T$). Again, for concreteness we have used a regression likelihood, but other likelihoods could also be used.

Now, we can integrate $\mathbf{F}_\ell$ out of the model, in which case, we get an equivalent generative model written solely in terms of Gram matrices (Fig. 1 bottom), with

$$\mathrm{P}\left(\mathbf{G}_\ell | \mathbf{G}_{\ell-1}\right) = \int d\mathbf{F}_\ell \ \mathrm{P}\left(\mathbf{G}_\ell | \mathbf{F}_\ell\right) \mathrm{P}\left(\mathbf{F}_\ell | \mathbf{G}_{\ell-1}\right), \tag{6}$$

and with the usual likelihood (e.g. Eq. 5c). This looks intractable (and indeed, in general it is intractable). However for DGPs, an analytic form is available. In particular, note the Gram matrix (Eq. 2) is the outer product of IID Gaussian distributed vectors (Eq. 1a). This matches the definition of the Wishart distribution (Gupta & Nagar, 2018), so we have,

$$\mathrm{P}\left(\mathbf{G}_\ell | \mathbf{G}_{\ell-1}\right) = \mathrm{Wishart}\left(\mathbf{G}_\ell; \tfrac{1}{N_\ell}\mathbf{K}(\mathbf{G}_{\ell-1}), N_\ell\right) \tag{7}$$

$$\log \mathrm{P}\left(\mathbf{G}_\ell | \mathbf{G}_{\ell-1}\right) = \tfrac{N_\ell - P - 1}{2}\log|\mathbf{G}_\ell| - \tfrac{N_\ell}{2}\log|\mathbf{K}(\mathbf{G}_{\ell-1})| - \tfrac{N_\ell}{2}\mathrm{Tr}\left(\mathbf{K}^{-1}(\mathbf{G}_{\ell-1})\mathbf{G}_\ell\right) + \mathrm{const}.$$

This distribution over Gram matrices is valid for DGPs of any width (though we need to be careful in the low-rank setting where $N_\ell < P$). We are going to leverage these Wishart distributions to understand the behaviour of the Gram matrices in the infinite width limit.

## 4.2 STANDARD INFINITE WIDTH LIMITS OF DGPS LACK REPRESENTATION LEARNING

We are now in a position to take a new viewpoint on the DGP analogue of standard NNGP results (Lee et al., 2017; Matthews et al., 2018; Hron et al., 2020; Pleiss & Cunningham, 2021). We can then evaluate the log-posterior for a model written only in terms of Gram matrices,

$$\log \mathrm{P}\left(\mathbf{G}_1, \ldots, \mathbf{G}_L | \mathbf{X}, \mathbf{Y}\right) = \log \mathrm{P}\left(\mathbf{Y} | \mathbf{G}_L\right) + \sum_{\ell=1}^{L} \log \mathrm{P}\left(\mathbf{G}_\ell | \mathbf{G}_{\ell-1}\right) + \mathrm{const}. \tag{8}$$

Then we take the limit of infinite width,

$$N_\ell = N\nu_\ell \qquad \text{for} \qquad \ell \in \{1, \ldots, L\} \qquad \text{with} \qquad N \to \infty. \tag{9}$$

This limit modifies $\log \mathrm{P}\left(\mathbf{G}_\ell | \mathbf{G}_{\ell-1}\right)$ (Eq. 7), but does not modify $\mathbf{G}_1, \ldots, \mathbf{G}_L$ in Eq. (8) as we get to choose the values of $\mathbf{G}_1, \ldots, \mathbf{G}_L$ at which to evaluate the log-posterior. Specifically, the log-prior, $\log \mathrm{P}\left(\mathbf{G}_\ell | \mathbf{G}_{\ell-1}\right)$ (Eq. 7), scales with $N_\ell$ and hence with $N$. To get a finite limit, we therefore need to divide by $N$,

$$\lim_{N \to \infty} \tfrac{1}{N} \log \mathrm{P}\left(\mathbf{G}_\ell | \mathbf{G}_{\ell-1}\right) = \tfrac{\nu_\ell}{2}\left(\log\left|\mathbf{K}^{-1}(\mathbf{G}_{\ell-1})\mathbf{G}_\ell\right| - \mathrm{Tr}\left(\mathbf{K}^{-1}(\mathbf{G}_{\ell-1})\mathbf{G}_\ell\right)\right) + \mathrm{const}$$

$$= -\nu_\ell \, \mathrm{D_{KL}}\left(\mathcal{N}\left(\mathbf{0}, \mathbf{G}_\ell\right)\|\mathcal{N}\left(\mathbf{0}, \mathbf{K}(\mathbf{G}_{\ell-1})\right)\right) + \mathrm{const}. \tag{10}$$

And remarkably this limit can be written as the KL-divergence between two multivariate Gaussians. In contrast, the log likelihood, $\log \mathrm{P}\left(\mathbf{Y} | \mathbf{G}_L\right)$, is constant wrt $N$ (Eq. 5c), so $\lim_{N \to \infty} \tfrac{1}{N}\log \mathrm{P}\left(\mathbf{Y} | \mathbf{G}_L\right) = 0$. The limiting log-posterior is thus,

$$\lim_{N \to \infty} \tfrac{1}{N} \log \mathrm{P}\left(\mathbf{G}_1, \ldots, \mathbf{G}_L | \mathbf{X}, \mathbf{Y}\right) = -\sum_{\ell=1}^{L}\nu_\ell \, \mathrm{D_{KL}}\left(\mathcal{N}\left(\mathbf{0}, \mathbf{G}_\ell\right)\|\mathcal{N}\left(\mathbf{0}, \mathbf{K}(\mathbf{G}_{\ell-1})\right)\right) + \mathrm{const}. \tag{11}$$

This form highlights that the log-posterior scales with $N$, so in the limit as $N \to \infty$, the posterior converges to a point distribution at the global maximum, denoted $\mathbf{G}_1^*, \ldots, \mathbf{G}_L^*$, (see Appendix C for a formal discussion of weak convergence),

$$\lim_{N \to \infty} \mathrm{P}\left(\mathbf{G}_1, \ldots, \mathbf{G}_L | \mathbf{X}, \mathbf{Y}\right) = \prod_{\ell=1}^{L}\delta\left(\mathbf{G}_\ell - \mathbf{G}_\ell^*\right). \tag{12}$$

Moreover, it is evident from the KL-divergence form for the log-posterior (Eq. 11) that the unique global maximum can be computed recursively as $\mathbf{G}_\ell^* = \mathbf{K}(\mathbf{G}_{\ell-1}^*)$, with e.g. $\mathbf{G}_0^* = \frac{1}{\nu_0}\mathbf{X}\mathbf{X}^T$. Thus, the limiting posterior over Gram matrices does not depend on the training targets, so there is no possibility of representation learning (Aitchison, 2020). This is deeply problematic as the successes of modern deep learning arise from flexibly learning good top-layer representations.

### 4.3 The Bayesian representation learning limit

In the previous section, we saw that standard infinite width limits eliminate representation learning because as $N \to \infty$ the log-prior terms, $\log \mathrm{P}\left(\mathbf{G}_\ell | \mathbf{G}_{\ell-1}\right)$, in Eq. (8) dominated the log-likelihood, $\mathrm{P}\left(\mathbf{Y}|\mathbf{G}_L\right)$, and the likelihood is the only term that depends on the labels. We therefore introduce the "Bayesian representation learning limit" which retains representation learning. The Bayesian representation learning limit sends the number of output features, $N_{L+1}$, to infinity as the layer-widths go to infinity,

$$N_\ell = N\,\nu_\ell \qquad \text{for} \qquad \ell \in \{1, \dots, L+1\} \qquad \text{with} \qquad N \to \infty. \qquad (13)$$

Importantly, the Bayesian representation learning limit gives a valid probabilistic model with a well-defined posterior, arising from the prior, (Eq. 6) and a likelihood which assumes each output channel is IID,

$$\mathrm{P}\left(\tilde{\mathbf{Y}}|\mathbf{G}_L\right) = \prod_{\lambda=1}^{N_{L+1}} \mathcal{N}\left(\tilde{\mathbf{y}}_\lambda; \mathbf{0}, \mathbf{K}(\mathbf{G}_L) + \sigma^2 \mathbf{I}\right). \qquad (14)$$

where $\tilde{\mathbf{Y}} \in \mathbb{R}^{P \times N_{L+1}}$ is infinite width (Eq. 13) whereas the usual DGP data, $\mathbf{Y} \in \mathbb{R}^{P \times \nu_{L+1}}$, is finite width. Of course, infinite-width data is unusual if not unheard-of. In practice, real data, $\mathbf{Y} \in \mathbb{R}^{P \times \nu_{L+1}}$, almost always has a finite number of features, $\nu_{L+1}$. How do we apply the DKM to such data? The answer is to define $\tilde{\mathbf{Y}}$ as $N$ copies of the underlying data, $\mathbf{Y}$, i.e. $\tilde{\mathbf{Y}} = \begin{pmatrix} \mathbf{Y} & \cdots & \mathbf{Y} \end{pmatrix}$. As each channel is assumed to be IID (Eq. 5c and 14) the likelihood is $N$ times larger,

$$\log \mathrm{P}\left(\tilde{\mathbf{Y}}|\mathbf{G}_L\right) = N \log \mathrm{P}\left(\mathbf{Y}|\mathbf{G}_L\right), \qquad (15)$$

The log-posterior in the Bayesian representation learning limit is very similar to the log-posterior in the standard limit (Eq. 11). The only difference is that the likelihood, $\log \mathrm{P}\left(\tilde{\mathbf{Y}}|\mathbf{G}_L\right)$ now scales with $N$, so it does not disappear as we take the limit, allowing us to retain representation learning,

$$\mathcal{L}(\mathbf{G}_1, \dots, \mathbf{G}_L) = \lim_{N \to \infty} \tfrac{1}{N} \log \mathrm{P}\left(\mathbf{G}_1, \dots, \mathbf{G}_L | \mathbf{X}, \tilde{\mathbf{Y}}\right) + \mathrm{const}, \qquad (16)$$
$$= \log \mathrm{P}\left(\mathbf{Y}|\mathbf{G}_L\right) - \sum_{\ell=1}^{L} \nu_\ell\, \mathrm{D}_{\mathrm{KL}}\left(\mathcal{N}\left(\mathbf{0}, \mathbf{G}_\ell\right) \| \mathcal{N}\left(\mathbf{0}, \mathbf{K}(\mathbf{G}_{\ell-1})\right)\right).$$

Here, we denote the limiting log-posterior as $\mathcal{L}(\mathbf{G}_1, \dots, \mathbf{G}_L)$, and this forms the DKM objective. Again, as long as the global maximum of the DKM objective is unique, the posterior is again a point distribution around that maximum (Eq. 12). Of course, the inclusion of the likelihood term means that the global optimum $\mathbf{G}_1^*, \dots, \mathbf{G}_L^*$ cannot be computed recursively, but instead we need to optimize, e.g. using gradient descent (see Sec. 4.7). Unlike in the standard limit (Eq. 11), it is no longer possible to guarantee uniqueness of the global maximum. We can nonetheless say that the posterior converges to a point distribution as long as the global maximum of $\mathcal{L}(\mathbf{G}_1, \dots, \mathbf{G}_L)$ is unique, (i.e. we can have any number of local maxima, as long as they all lie below the unique global maximum). We do expect the global maximum to be unique in most practical settings: we know the maximum is unique when the prior dominates (Eq. 11), in Appendix J, we prove uniqueness for linear models, and in Appendix K, we give a number of experiments in nonlinear models in which optimizing from very different initializations found the same global maximum, indicating uniqueness in practical settings.

### 4.4 The exact DGP posterior over features is multivariate Gaussian

Above, we noted that the DGP posterior over Gram matrices in the Bayesian representation learning limit is a point distribution, as long as the DKM objective has a unique global maximum. Remarkably, in this setting, the corresponding posterior over features is multivariate Gaussian (see Appendix D for the full derivation),

$$\mathrm{P}\left(\mathbf{f}_\lambda^\ell | \mathbf{X}, \mathbf{y}\right) = \mathcal{N}\left(\mathbf{f}_\lambda^\ell; \mathbf{0}, \mathbf{G}_\ell^*\right) \qquad (17)$$

While such a simple result might initially seem remarkable, it should not surprise us too much. In particular, the prior is Gaussian (Eq. 1). In addition, in Fig. 1 (middle), we saw that the next layer features depend on the current layer features only through the Gram matrices, which are just the raw second moment of the features, Eq. (2). Thus, in effect the likelihood only constrains the raw second moments of the features. Critically, that constraints on the raw second moment are tightly connected to Gaussian distributions: under the MaxEnt framework, a Gaussian distribution arises by maximizing the entropy under constraints on the raw second moment of the features (Jaynes, 2003).

Thus it is entirely plausible that a Gaussian prior combined with a likelihood that "constrains" the raw second moment would give rise to Gaussian posteriors (though of course this is not a proof; see Appendix D for the full derivation).

Finally, note that we appear to use $\mathbf{G}_\ell$ or $\mathbf{G}_\ell^*$ in two separate senses: as $\frac{1}{N_\ell}\mathbf{F}_\ell\mathbf{F}_\ell^T$ in Eq. (2) and as the posterior covariance in the Bayesian representation learning limit (Eq. 17). In the infinite limit, these two uses are consistent. In particular, consider the value of $\mathbf{G}_\ell$ defined by Eq. (2) under the posterior,

$$\mathbf{G}_\ell = \lim_{N\to\infty} \tfrac{1}{N_\ell}\sum_{\lambda=1}^{N_\ell}\mathbf{f}_\lambda^\ell(\mathbf{f}_\lambda^\ell)^T = \mathbb{E}_{\mathrm{P}(\mathbf{f}_\lambda^\ell|\mathbf{X},\mathbf{y})}\left[\mathbf{f}_\lambda^\ell(\mathbf{f}_\lambda^\ell)^T\right] = \mathbf{G}_\ell^*. \tag{18}$$

The second equality arises by noticing that we are computing the average of infinitely many terms, $\mathbf{f}_\lambda^\ell(\mathbf{f}_\lambda^\ell)^T$, which are IID under the true posterior (Eq. 17), so we can apply the law of large numbers, and the final expectation arises by computing moments under Eq. (17).

### 4.5 THE DKM OBJECTIVE GIVES INTUITION FOR REPRESENTATION LEARNING

The form for the DKM objective in Eq. (16) gives a strong intuition for how representation learning occurs in deep networks. In particular, the likelihood, $\log\mathrm{P}\left(\mathbf{Y}|\mathbf{G}_L\right)$, encourages the model to find a representation giving good performance on the training data. At the same time, the KL-divergence terms keep the posterior over features, $\mathcal{N}\left(\mathbf{0},\mathbf{G}_\ell\right)$, (Eq. 17) close to the prior $\mathcal{N}\left(\mathbf{0},\mathbf{K}(\mathbf{G}_{\ell-1})\right)$ (Eq. 1a). This encourages the optimized representations, $\mathbf{G}_\ell$, to lie close to their value under the standard infinite-width limit, $\mathbf{K}(\mathbf{G}_{\ell-1})$. We could use any form for the likelihood including classification and regression, but to understand how the likelihood interacts with the other KL-divergence terms, it is easiest to consider regression (Eq. 5c), as this log-likelihood can also be written as a KL-divergence,

$$\log\mathrm{P}\left(\mathbf{Y}|\mathbf{G}_L\right) = -\nu_{L+1}\,\mathrm{D}_{\mathrm{KL}}\left(\mathcal{N}\left(\mathbf{0},\mathbf{G}_{L+1}\right)\big\|\mathcal{N}\left(\mathbf{0},\mathbf{K}(\mathbf{G}_L)+\sigma^2\mathbf{I}\right)\right) + \mathrm{const} \tag{19}$$

Thus, the likelihood encourages $\mathbf{K}(\mathbf{G}_L)+\sigma^2\mathbf{I}$ to be close to the covariance of the data, $\mathbf{G}_{L+1} = \frac{1}{\nu_{L+1}}\mathbf{Y}\mathbf{Y}^T$, while the DGP prior terms encourage all $\mathbf{G}_\ell$ to lie close to $\mathbf{K}(\mathbf{G}_{\ell-1})$. In combination, we would expect the optimal Gram matrices to "interpolate" between the input kernel, $\mathbf{G}_0 = \frac{1}{\nu_0}\mathbf{X}\mathbf{X}^T$ and the output kernel, $\mathbf{G}_{L+1}$.

To make the notion of interpolation explicit, we consider $\sigma^2 = 0$ with a linear kernel, $\mathbf{K}(\mathbf{G}_{\ell-1}) = \mathbf{G}_{\ell-1}$, so named because it corresponds to a linear neural network layer. With this kernel and with all $\nu_\ell = \nu$, there is an analytic solution for the (unique) optimum of the DKM objective (Appendix J.1),

$$\mathbf{G}_\ell^* = \mathbf{G}_0\left(\mathbf{G}_0^{-1}\mathbf{G}_{L+1}\right)^{\ell/(L+1)}, \tag{20}$$

which explicitly geometrically interpolates between $\mathbf{G}_0$ and $\mathbf{G}_{L+1}$. Of course, this discussion was primarily for DGPs, but the exact same intuitions hold for BNNs, in that maximizing the DKM objective finds a sequence of Gram matrices, $\mathbf{G}_1^*,\dots,\mathbf{G}_L^*$ that interpolate between the input kernel, $\mathbf{G}_0$ and the output kernel, $\mathbf{G}_{L+1}$. The only difference is in details of $\mathrm{P}\left(\mathbf{G}_\ell|\mathbf{G}_{\ell-1}\right)$, and specifically as slight differences in the KL-divergence terms (see below).

### 4.6 THE DKM OBJECTIVE MIRRORS REPRESENTATION LEARNING IN FINITE NETWORKS

Here, we confirm that the optimizing DKM objective for an infinite network matches doing inference in wide but finite-width networks using Langevin sampling (see Appendix F for details).

We began by looking at DGPs, and confirming that the posterior marginals are Gaussian (Eq. 17; Fig. 3ab). Then, we confirmed that the representations match closely for infinite-width DKMs (Fig. 2 top and bottom rows) and finite-width DGPs (Fig. 2 middle two rows), both at initialization (Fig. 2 top two rows) and after training to convergence (Fig. 2 bottom two rows). Note that the first column, $\mathbf{K}_0$ is a squared exponential kernel applied to the input data, and $\mathbf{G}_3 = \mathbf{y}\mathbf{y}^T$ is the output Gram matrix (in this case, there is only one output feature).

To confirm that the match improves as the DGP gets wider, we considered the RMSE between elements of the Gram matrices for networks of different widths (x-axis) for different UCI datasets (columns) and different numbers of layers (top row is one-layer, bottom row is two-layers; Fig. 3c). In most cases, we found a good match as long as the width was at least 128, which is around the width of typical fully connected neural network, but is a little larger than typical DGP widths (e.g. Damianou & Lawrence, 2013; Salimbeni & Deisenroth, 2017).

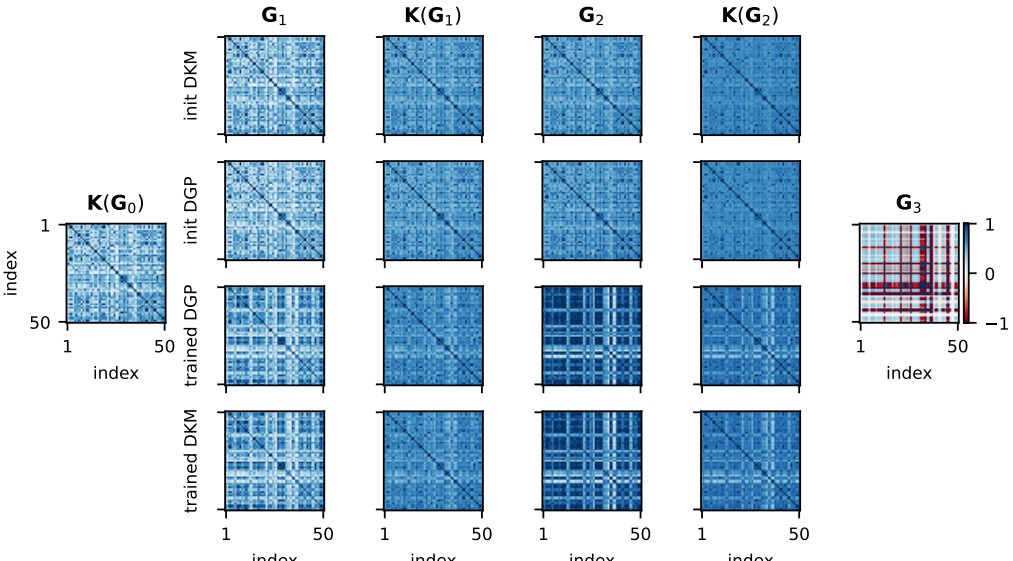

Figure 2: A two hidden layer DGP with 1024 units per hidden layer and DKM with squared exponential kernels match closely. The data was the first 50 datapoints of the yacht dataset. The first column, $\mathbf{K}_0$ is a fixed squared exponential kernel applied to the inputs, and the last column, $\mathbf{G}_3 = \mathbf{yy}^T$ is the fixed output Gram matrix. The first row is the DKM initialization at the prior Gram matrices and kernels, and the second row is the DGP, which is initialized by sampling from the prior. As expected, the finite width DGP prior closely matches the infinite-width DKM initialization, which corresponds to the standard infinite width limit. The third row is the Gram matrices and kernels for the trained DGP, which has changed dramatically relative to its initialization (second row) in order to better fit the data. The fourth row is the Gram matrices and kernels for the optimized DKM, which closely matches those for the trained DGP.

### 4.7 THE SPARSE DEEP KERNEL MACHINE AS A DEEP GENERALISATION OF KERNEL METHODS

DGPs in the representation learning limit constitute a deep generalisation of kernel methods, with a very flexible learned kernel, which we call the deep kernel machine (DKM; which was introduced earlier just in the context of the objective). Here, we design a sparse DKM, inspired by sparse methods for DGPs (Damianou & Lawrence, 2013; Salimbeni & Deisenroth, 2017) (Appendix L). The sparse DKM scales linearly in the number of datapoints, $P$, as opposed to cubic scaling of the plain DKM (similar to the cubic scaling in most naive kernel methods).

We compared DKMs (Eq. 16) and MAP over features (Sec. E) for DGPs. In addition, we considered a baseline, which was a standard, shallow kernel method mirroring the structure of the deep kernel machine but where the only flexibility comes from the hyperparameters. Formally, this model can be obtained by setting, $\mathbf{G}_\ell = \mathbf{K}(\mathbf{G}_{\ell-1})$ and is denoted "Kernel Hyper" in Table 1. We applied these methods to UCI datasets (Gal & Ghahramani, 2016) using a two hidden layer architecture, with a kernel inspired by DGP skip-connections, $\mathbf{K}(\mathbf{G}_\ell) = w_1^\ell \mathbf{G}_\ell + w_2^\ell \mathbf{K}_{\mathrm{sqexp}}(\mathbf{G}_\ell)$. Here, $w_1^\ell$, $w_2^\ell$ and $\sigma$ are hyperparameters, and $\mathbf{K}_{\mathrm{sqexp}}(\mathbf{G}_\ell)$ is a squared-exponential kernel.

We used 300 inducing points fixed to a random subset of the training data and not optimised during training. We used the Adam optimizer with a learning rate of 0.001, full-batch gradients and 5000 iterations for smaller datasets and 1000 iterations for larger datasets (kin8nm, naval and protein).

We found that the deep kernel machine objective gave better performance than MAP, or the hyperparameter optimization baseline (Tab. 1). Note that these numbers are not directly comparable to those in the deep GP literature (Salimbeni & Deisenroth, 2017), as deep GPs have a full posterior so offer excellent protection against overfitting, while DKMs give only a point estimate.

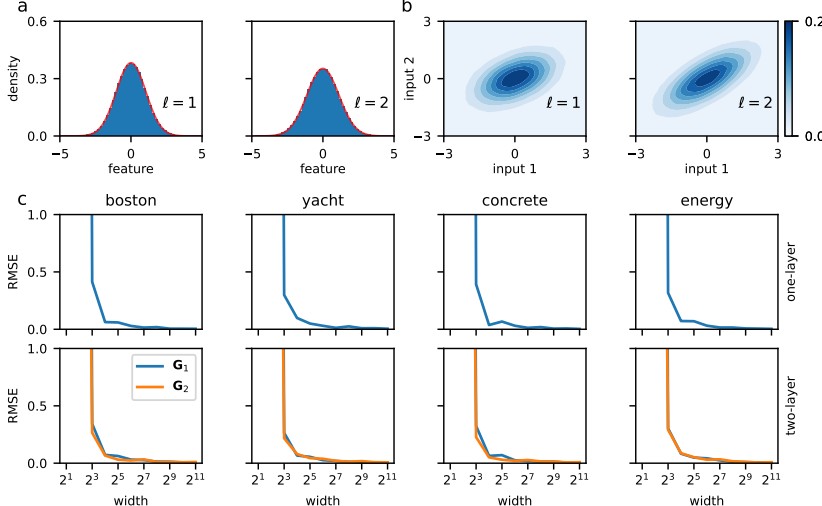

Figure 3: Wide DGP posteriors converge to the DKM. Here, we trained DGPs with Langevin sampling (see Appendix F), and compared to a trained DKM. **a** Marginal distribution over features for one input datapoint for a two-layer DGP trained on a subset of yacht. We used a width of $N_{1...L} = 1024$ and $\nu_{1...L} = 5$ in all plots to ensure that the data had a strong effect on the learned representations. The marginals (blue histogram) are very close to Gaussian (the red line shows the closest fitted Gaussian). Remember that the true posterior over features is IID (Eq. 31), so each column aggregates the distribution over features (and over 10 parallel chains with 100 samples from each chain) for a single input datapoint. **b** The 2D marginal distributions for the same DGP for two input points (horizontal and vertical axes). **c** Element-wise RMSE (normalized Frobenius distance) between Gram matrices from a trained DKM compared to trained DGPs of increasing width. The DGP Gram matrices converge to the DKM solution as width becomes larger.

Table 1: RMSE for inducing point methods. (Equal) best methods are displayed in bold. Error bars give two stderrs for a paired tests, which uses differences in performance between that method and best method, (so there are no meaningful error bars on the best performing method itself). The MAP objective was numerically unstable and thus did not run to completion on the boston dataset.

| dataset | P | Kernel Hyper | MAP | $\mathcal{L}$ |
|---|---|---|---|---|
| boston | 506 | $\mathbf{4.41 \pm 0.31}$ | — | $\mathbf{4.35 \pm 0.51}$ |
| concrete | 1,030 | $5.38 \pm 0.098$ | $5.60 \pm 0.15$ | $\mathbf{5.10}$ |
| energy | 768 | $0.83 \pm 0.076$ | $0.73 \pm 0.049$ | $\mathbf{0.47}$ |
| kin8nm | 8,192 | $(7.3 \pm 0.06) \cdot 10^{-2}$ | $(7.4 \pm 0.05) \cdot 10^{-2}$ | $\mathbf{6.6 \cdot 10^{-2}}$ |
| naval | 11,934 | $(6.4 \pm 0.6) \cdot 10^{-4}$ | $(5.4 \pm 0.5) \cdot 10^{-4}$ | $\mathbf{4.6 \cdot 10^{-4}}$ |
| power | 9,568 | $3.81 \pm 0.091$ | $3.73 \pm 0.14$ | $\mathbf{3.58}$ |
| protein | 45,730 | $4.21 \pm 0.029$ | $4.30 \pm 0.033$ | $\mathbf{4.10}$ |
| wine | 1,599 | $0.68 \pm 0.0084$ | $0.66 \pm 0.0067$ | $\mathbf{0.64}$ |
| yacht | 308 | $0.94 \pm 0.058$ | $1.14 \pm 0.077$ | $\mathbf{0.58}$ |

## 5 CONCLUSION

We introduced the Bayesian representation learning limit, a new infinite-width limit for BNNs and DGPs that retains representation learning. Representation learning in this limit is described by the intuitive DKM objective, which is composed of a log-likelihood describing performance on the task (e.g. classification or regression) and a sum of KL-divergences keeping representations at every layer close to those under the infinite-width prior. For DGPs, the exact posteriors are IID across features and are multivariate Gaussian, with covariances given by optimizing the DKM objective. Empirically, we found that the distribution over features and representations matched those in wide by finite DGPs. We argued that DGPs in the Bayesian representation learning limit form a new

class of practical deep kernel method: DKMs. We introduce sparse DKMs, which scale linearly in the number of datapoints. Finally, we give the extension for BNNs where the exact posteriors are intractable so must be approximated.

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

## A   BAYESIAN NEURAL NETWORK EXTENSION

Consider a neural network of the form,

$$\mathbf{F}_1 = \mathbf{X}\mathbf{W}_0 \tag{21a}$$

$$\mathbf{F}_\ell = \phi(\mathbf{F}_{\ell-1})\mathbf{W}_{\ell-1} \quad \text{for } \ell \in \{2, \dots, L+1\} \tag{21b}$$

$$W_{\lambda\mu}^\ell \sim \mathcal{N}\left(0, \tfrac{1}{N_\ell}\right) \qquad W_{\lambda\mu}^0 \sim \mathcal{N}\left(0, \tfrac{1}{\nu_0}\right) \tag{21c}$$

where $\mathbf{W}_0 \in \mathbb{R}^{\nu_0 \times N_1}$, $\mathbf{W}_\ell \in \mathbb{R}^{N_\ell \times N_{\ell+1}}$ and $\mathbf{W}_{L+1} \in \mathbb{R}^{N_L \times \nu_{L+1}}$ are weight matrices with independent Gaussian priors and $\phi$ is the usually pointwise nonlinearity.

In principle, we could integrate out the distribution over $\mathbf{W}_\ell$ to find $\mathrm{P}\left(\mathbf{F}_\ell | \mathbf{F}_{\ell-1}\right)$

$$\mathrm{P}\left(\mathbf{F}_\ell | \mathbf{F}_{\ell-1}\right) = \int d\mathbf{W}_\ell \, \mathrm{P}\left(\mathbf{W}_\ell\right) \delta\left(\mathbf{F}_\ell - \phi(\mathbf{F}_{\ell-1})\mathbf{W}_{\ell-1}\right), \tag{22}$$

where $\delta$ is the Dirac delta. In practice, it is much easier to note that conditioned on $\mathbf{F}_{\ell-1}$, the random variables interest, $\mathbf{F}_\ell$ are a linear combination of Gaussian distributed random variables, $\mathbf{W}_\ell$. Thus, $\mathbf{F}_\ell$ are themselves Gaussian, and this Gaussian is completely characterised by its mean and variance. We begin by writing the feature vectors, $\mathbf{f}_\lambda^\ell$ in terms of weight vectors, $\mathbf{w}_\lambda^\ell$,

$$\mathbf{f}_\lambda^\ell = \phi(\mathbf{F}_{\ell-1})\mathbf{w}_\lambda^\ell. \tag{23}$$

As the prior over weight vectors is IID, the prior over features, conditioned on $\mathbf{F}_{\ell-1}$), is also IID,

$$\mathrm{P}\left(\mathbf{W}\right) = \prod_{\lambda=1}^{N_\ell} \mathrm{P}\left(\mathbf{w}_\lambda^\ell\right) = \prod_{\lambda=1}^{N_\ell} \mathcal{N}\left(\mathbf{w}_\lambda^\ell; \mathbf{0}, \tfrac{1}{N_{\ell-1}}\mathbf{I}\right), \tag{24}$$

$$\mathrm{P}\left(\mathbf{F}_\ell | \mathbf{F}_{\ell-1}\right) = \prod_{\lambda=1}^{N_\ell} \mathrm{P}\left(\mathbf{f}_\lambda^\ell | \mathbf{F}_{\ell-1}\right). \tag{25}$$

The mean of $\mathbf{f}_\lambda^\ell$ conditioned on $\mathbf{F}_{\ell-1}$ is $\mathbf{0}$,

$$\mathbb{E}\left[\mathbf{f}_\lambda^\ell | \mathbf{F}_{\ell-1}\right] = \mathbb{E}\left[\phi(\mathbf{F}_{\ell-1})\mathbf{w}_\lambda^\ell | \mathbf{F}_{\ell-1}\right] = \phi(\mathbf{F}_{\ell-1})\mathbb{E}\left[\mathbf{w}_\lambda^\ell | \mathbf{F}_{\ell-1}\right] = \phi(\mathbf{F}_{\ell-1})\mathbb{E}\left[\mathbf{w}_\lambda^\ell\right] = \mathbf{0}. \tag{26}$$

The covariance of $\mathbf{f}_\lambda^\ell$ conditioned on $\mathbf{F}_{\ell-1}$ is,

$$\begin{aligned}
\mathbb{E}\left[\mathbf{f}_\lambda^\ell \left(\mathbf{f}_\lambda^\ell\right)^T | \mathbf{F}_{\ell-1}\right] &= \mathbb{E}\left[\phi(\mathbf{F}_{\ell-1})\mathbf{w}_\lambda^\ell \left(\phi(\mathbf{F}_{\ell-1})\mathbf{w}_\lambda^\ell\right)^T | \mathbf{F}_{\ell-1}\right] \\
&= \phi(\mathbf{F}_{\ell-1})\mathbb{E}\left[\mathbf{w}_\lambda^\ell(\mathbf{w}_\lambda^\ell)^T\right]\phi^T(\mathbf{F}_{\ell-1}) \\
&= \tfrac{1}{N_{\ell-1}}\phi(\mathbf{F}_{\ell-1})\phi^T(\mathbf{F}_{\ell-1}) \tag{27}
\end{aligned}$$

This mean and variance imply that Eq. (1) captures the BNN prior, as long as we choose $\mathbf{K}_{\mathrm{BNN}}(\cdot)$ and $\mathbf{G}_{\mathrm{BNN}}(\cdot)$ such that,

$$\mathbf{K}_{\mathrm{BNN}}(\mathbf{G}_{\mathrm{BNN}}(\mathbf{F}_{\ell-1})) = \tfrac{1}{N_{\ell-1}}\textstyle\sum_{\lambda=1}^{N_{\ell-1}} \phi(\mathbf{f}_\lambda^{\ell-1})\phi^T(\mathbf{f}_\lambda^{\ell-1}), \tag{28}$$

Specifically, we choose the kernel function, $\mathbf{K}_{\mathrm{BNN}}(\cdot)$ to be the identity function, and $\mathbf{G}_{\mathrm{BNN}}(\cdot)$ to be the same outer product as in the main text for DGPs (Eq. 2), except where we have applied the NN nonlinearity,

$$\mathbf{K}_{\mathrm{BNN}}(\mathbf{G}_{\ell-1}) = \mathbf{G}_{\ell-1}, \tag{29}$$

$$\mathbf{G}_{\mathrm{BNN}}(\mathbf{F}_{\ell-1}) = \tfrac{1}{N_{\ell-1}}\textstyle\sum_{\lambda=1}^{N_{\ell-1}} \phi(\mathbf{f}_\lambda^{\ell-1})\phi^T(\mathbf{f}_\lambda^{\ell-1}). \tag{30}$$

This form retains the average-outer-product form for $\mathbf{G}_{\mathrm{BNN}}(\cdot)$, which is important for our derivations.

Now, Eq. (16) only gave the DKM objective for DGPs. To get a more general form, we need to consider the implied posteriors over features. This posterior is IID over features (Appendix D.1), and for DGPs, it is multivariate Gaussian (Appendix D.2),

$$\mathrm{P}\left(\mathbf{F}_\ell | \mathbf{G}_{\ell-1}, \mathbf{G}_\ell\right) = \textstyle\prod_{\lambda=1}^{N^\ell} \mathrm{P}\left(\mathbf{f}_\lambda^\ell | \mathbf{G}_{\ell-1}, \mathbf{G}_\ell\right) \underset{\text{for DGPs}}{=} \textstyle\prod_{\lambda=1}^{N^\ell} \mathcal{N}\left(\mathbf{f}_\lambda^\ell; \mathbf{0}, \mathbf{G}_\ell\right). \tag{31}$$

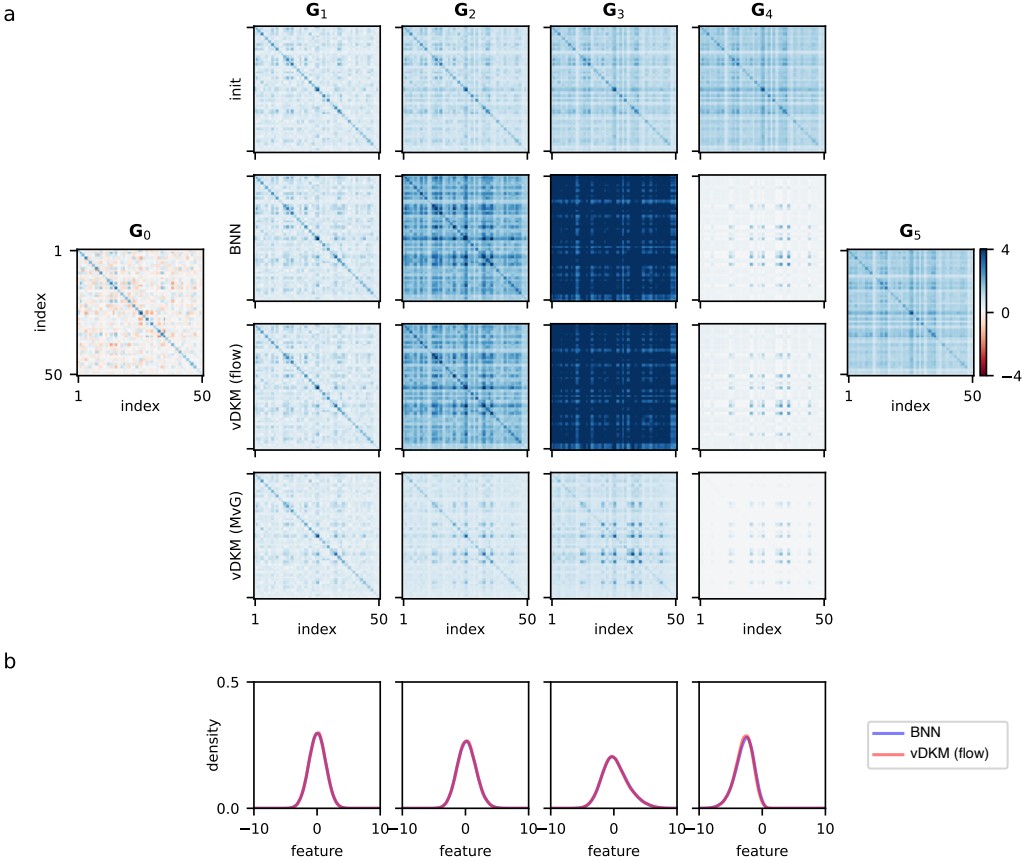

Figure 4: The variational DKM closely matches the BNN true posterior obtained with Langevin sampling. **a** Comparison of Gram matrices. The first two rows show Gram matrices for BNN, with the first row being a random initialization, and the second row being the posterior. The last two rows show the Gram matrices from variational DKMs with a flow approximate posterior (third row) and a multivariate Gaussian approximate posterior (fourth row). In optimizing the variational DKM, we used Eq. (34) with $2^{16}$ Monte-Carlo samples. The Gram matrices for the flow posterior (third row) closely match those from the BNN posterior (second row), while those for a multivariate Gaussian approximate posterior (fourth row) do not match. **b** Marginal distributions over features at each layer for one input datapoint estimated using kernel density estimation. The note that the BNN (blue line) marginals are non-Gaussian, but the variational DKM with a flow posterior (red line) is capable of capturing this non-Gaussianity.

Now, we can see that Eq. (16) is a specific example of a general expression. In particular, note that the distribution on the left of the KL-divergence in Eq. (16) is the DGP posterior over features (Eq. 31). Thus, the DKM objective can alternatively be written,

$$\mathcal{L}(\mathbf{G}_1, \ldots, \mathbf{G}_L) = \log \mathrm{P}\left(\mathbf{Y}|\mathbf{G}_L\right) - \sum_{\ell=1}^{L} \nu_\ell \, \mathrm{D}_{\mathrm{KL}}\left(\mathrm{P}\left(\mathbf{f}_\lambda^\ell|\mathbf{G}_{\ell-1}, \mathbf{G}_\ell\right)\big\|\mathcal{N}\left(\mathbf{0}, \mathbf{K}(\mathbf{G}_{\ell-1})\right)\right), \quad (32)$$

and this form holds for both BNNs and DGPs (Appendix D.3). As in DGPs, the log-posterior is $N$ times $\mathcal{L}(\mathbf{G}_1, \ldots, \mathbf{G}_L)$ (Eq. 16), so as $N$ is taken to infinity, the posterior for all models becomes a point distribution (Eq. 12) if $\mathcal{L}(\mathbf{G}_1, \ldots, \mathbf{G}_L)$ has a unique global maximum.

In practice, the true posteriors required to evaluate Eq. (32) are intractable for BNNs, raising the question of how to develop accurate approximations for BNNs. We develop a variational DKM (vDKM) by taking inspiration from variational inference (Jordan et al., 1999; Blei et al., 2017) (Appendix D.4). Of course, variational inference is usually impossible in infinite width models, because it is impossible to work with infinitely large latent variables. Our key insight is that as the true posterior factorises across features (Appendix D.1), we can work with the approximate posterior

over only a single feature vector, $Q_{\theta_\ell}\left(\mathbf{f}_\lambda^\ell\right)$, where $\theta_\ell$ are the parameters and $\mathbf{f}_\lambda^\ell \in \mathbb{R}^P$ is finite. This approach allows us to define a vDKM objective, which bounds the true DKM objective,

$$\mathcal{L}(\mathbf{G}_\theta(\theta_1), \ldots, \mathbf{G}_\theta(\theta_L)) \geq \mathcal{L}_{\mathrm{V}}(\theta_1, \ldots, \theta_L), \tag{33}$$

$$\mathcal{L}_{\mathrm{V}}(\theta_1, \ldots, \theta_L) = \log \mathrm{P}\left(\mathbf{Y}|\mathbf{G}_\theta(\theta_L)\right) - \sum_{\ell=1}^{L} \nu_\ell \, \mathrm{D}_{\mathrm{KL}}\left(Q_{\theta_\ell}\left(\mathbf{f}_\lambda^\ell\right)\big\|\mathcal{N}\left(\mathbf{0}, \mathbf{K}(\mathbf{G}_\theta(\theta_{\ell-1}))\right)\right)$$

with equality when the approximate posteriors, $Q_{\theta_\ell}\left(\mathbf{f}_\lambda^\ell\right)$, equal the true posteriors, $\mathrm{P}\left(\mathbf{f}_\lambda^\ell|\mathbf{G}_{\ell-1}, \mathbf{G}_\ell\right)$. The only subtlety here is that it is practically difficult to design flexible approximate posteriors $Q_{\theta_\ell}\left(\mathbf{f}_\lambda^\ell\right)$ where we explicitly specify and optimize the Gram matrices. Instead we optimize general approximate posterior parameters, $\theta$, and compute the implied Gram matrices,

$$\mathbf{G}_\theta(\theta_\ell) = \frac{1}{N_\ell} \lim_{N_\ell \to \infty} \sum_{\lambda=1}^{N_\ell} \phi(\mathbf{f}_\lambda^\ell) \phi^T(\mathbf{f}_\lambda^\ell) = \mathbb{E}_{Q_{\theta_\ell}(\mathbf{f}_\lambda^\ell)}\left[\phi(\mathbf{f}_\lambda^\ell)\phi^T(\mathbf{f}_\lambda^\ell)\right]. \tag{34}$$

where $\mathbf{f}_\lambda^\ell$ are sampled from $Q_{\theta_\ell}\left(\mathbf{f}_\lambda^\ell\right)$, and the second equality arises from the law of large numbers. We can compute the Gram matrix analytically in simple cases (such as a multivariate Gaussian), but in general we can always estimate the Gram matrix using a Monte-Carlo estimate of Eq. (34).

Finally, we checked that the vDKM objective closely matched the posterior under neural networks. This is a bit more involved, as the marginal distributions over features are no longer Gaussian (Fig. 4b). To capture these non-Gaussian marginals, we used a simple normalizing flow. In particular, we first sampled $\mathbf{z}_\lambda^\ell \sim \mathcal{N}\left(\boldsymbol{\mu}_\ell, \boldsymbol{\Sigma}_\ell\right)$ from a multivariate Gaussian with a learned mean, $\boldsymbol{\mu}_\ell$, and covariance, $\boldsymbol{\Sigma}_\ell$ then we obtained features, $\mathbf{f}_\lambda^\ell = f(\mathbf{z}_\lambda^\ell)$, by passing $\mathbf{z}_\lambda^\ell$ through $f$, a learned pointwise function parameterised as in a neural spline flow (Durkan et al., 2019). The resulting distribution is a high-dimensional Gaussian copula (e.g. Cai & Zhang, 2018). As shown in Fig. 4, vDKM with multivariate Gaussian (MvG) approximate posterior cannot match the Gram matrices learned by BNN (Fig. 4a), while vDKM with flow is able to capture the non-Gaussian marginals (Fig. 4b) and thus match the learned Gram matrices with BNN.

## B  GENERAL LIKELIHOODS THAT DEPEND ONLY ON GRAM MATRICES

We consider likelihoods which depend only on the top-layer Gram matrix, $\mathbf{G}_L$,

$$\mathrm{P}\left(\mathbf{Y}|\mathbf{G}_L\right) = \int d\mathbf{F}_{L+1}\,\mathrm{P}\left(\mathbf{Y}|\mathbf{F}_{L+1}\right)\mathrm{P}\left(\mathbf{F}_{L+1}|\mathbf{G}_L\right) \tag{35}$$

where,

$$\mathrm{P}\left(\mathbf{F}_{L+1}|\mathbf{G}_L\right) = \prod_{\lambda=1}^{N_{L+1}} \mathcal{N}\left(\mathbf{f}_\lambda^{L+1}; \mathbf{0}, \mathbf{K}(\mathbf{G}_L)\right). \tag{36}$$

This family of likelihoods captures regression,

$$\mathrm{P}\left(\mathbf{y}_\lambda|\mathbf{f}_\lambda^{L+1}\right) = \mathcal{N}\left(\mathbf{y}_\lambda^{L+1}; \mathbf{f}_\lambda^{L+1}, \sigma^2\mathbf{I}\right), \tag{37}$$

(which is equivalent to the model used in the main text Eq. 1b) and e.g. classification,

$$\mathrm{P}\left(\mathbf{y}|\mathbf{F}\right) = \mathrm{Categorical}\left(\mathbf{y}; \mathrm{softmax}\left(\mathbf{F}_{L+1}\right)\right), \tag{38}$$

amoung many others.

## C  WEAK CONVERGENCE

Here, we give a formal argument for weak convergence of the DGP posterior over Gram matrices to a point distribution in the limit as $N \to \infty$,

$$\mathrm{P}_N\left(\mathbf{G}_1, \ldots, \mathbf{G}_L|\mathbf{X}, \tilde{\mathbf{Y}}\right) \xrightarrow{d} \prod_{\ell=1}^{L} \delta(\mathbf{G}_\ell - \mathbf{G}_\ell^*) \tag{39}$$

where we have included $N$ in the subscript of the probability distribution as a reminder that this distribution depends on the width. By the Portmanteau theorem, weak convergence is established if all expectations of bounded continuous functions, $f$, converge

$$\lim_{N \to \infty} \mathbb{E}_{P_N(\mathbf{G}_1, \ldots, \mathbf{G}_L | \mathbf{X}, \tilde{\mathbf{Y}})} [f(\mathbf{G}_1, \ldots, \mathbf{G}_L)] = f(\mathbf{G}_1^*, \ldots, \mathbf{G}_L^*). \tag{40}$$

To show this in a reasonably general setting (which the DGP posterior is a special case of), we consider an unnormalized probability density of the form $h(g)e^{N\mathcal{L}(g)}$, and compute the moment as,

$$\mathbb{E}[f(g)] = \frac{\int_{\mathcal{G}} dg \, f(g)h(g)e^{N\mathcal{L}(g)}}{\int_{\mathcal{G}} dg \, h(g)e^{N\mathcal{L}(g)}} \tag{41}$$

where $g = (\mathbf{G}_1, \ldots, \mathbf{G}_L)$ is all $L$ positive semi-definite matrices, $\mathbf{G}_\ell$. Thus, $g \in \mathcal{G}$, where $\mathcal{G}$ is a convex set.

We consider the superlevel set $A(\Delta) = \{g | \mathcal{L}(g) \geq \mathcal{L}(g^*) - \Delta\}$, where $g^*$ is the unique global optimum. We select out a small region, $A(\Delta)$, surrounding the global maximum, and compute the integral as,

$$\mathbb{E}[f(g)] = \frac{\int_{A(\Delta)} dg \, f(g)h(g)e^{N\mathcal{L}(g)} + \int_{\mathcal{G} \setminus A(\Delta)} dg \, f(g)h(g)e^{N\mathcal{L}(g)}}{\int_{A(\Delta)} dg \, h(g)e^{N\mathcal{L}(g)} + \int_{\mathcal{G} \setminus A(\Delta)} dg \, h(g)e^{N\mathcal{L}(g)}} \tag{42}$$

And divide the numerator and denominator by $\int_{A(\Delta)} dg \, h(g)e^{N\mathcal{L}(g)}$,

$$\mathbb{E}[f(g)] = \frac{\frac{\int_{A(\Delta)} dg \, f(g)h(g)e^{N\mathcal{L}(g)}}{\int_{A(\Delta)} dg \, h(g)e^{N\mathcal{L}(g)}} + \frac{\int_{\mathcal{G} \setminus A(\Delta)} dg \, f(g)h(g)e^{N\mathcal{L}(g)}}{\int_{A(\Delta)} dg \, h(g)e^{N\mathcal{L}(g)}}}{1 + \frac{\int_{\mathcal{G} \setminus A(\Delta)} dg \, h(g)e^{N\mathcal{L}(g)}}{\int_{A(\Delta)} dg \, h(g)e^{N\mathcal{L}(g)}}} \tag{43}$$

Now, we deal with each term separately. The ratio in the denominator can be lower-bounded by zero, and upper bounded by considering a smaller superlevel set, $A(\Delta/2)$, in the denominator,

$$0 \leq \frac{\int_{\mathcal{G} \setminus A(\Delta)} dg \, h(g)e^{N\mathcal{L}(g)}}{\int_{A(\Delta)} dg \, h(g)e^{N\mathcal{L}(g)}} \leq \frac{\int_{\mathcal{G} \setminus A(\Delta)} dg \, h(g)e^{N\mathcal{L}(g)}}{\int_{A(\Delta/2)} dg \, h(g)e^{N\mathcal{L}(g)}}$$

$$\leq \frac{e^{N(\mathcal{L}(g^*) - \Delta)} \int_{\mathcal{G} \setminus A(\Delta)} dg \, h(g)}{e^{N(\mathcal{L}(g^*) - \Delta/2)} \int_{A(\Delta/2)} dg \, h(g)}$$

$$= \frac{\int_{\mathcal{G} \setminus A(\Delta)} dg \, h(g)}{\int_{A(\Delta/2)} dg \, h(g)} e^{-N\Delta/2} \tag{44}$$

The upper bound converges to zero (as $h(g)$ is independent of $N$), and therefore by the sandwich theorem the ratio of interest also tends to zero.

The second ratio in the numerator can be rewritten as,

$$\frac{\int_{\mathcal{G} \setminus A(\Delta)} dg \, f(g)h(g)e^{N\mathcal{L}(g)}}{\int_{A(\Delta)} dg \, h(g)e^{N\mathcal{L}(g)}} = \frac{\int_{\mathcal{G} \setminus A(\Delta)} dg \, f(g)h(g)e^{N\mathcal{L}(g)}}{\int_{\mathcal{G} \setminus A(\Delta)} dg \, h(g)e^{N\mathcal{L}(g)}} \frac{\int_{\mathcal{G} \setminus A(\Delta)} dg \, h(g)e^{N\mathcal{L}(g)}}{\int_{A(\Delta)} dg \, h(g)e^{N\mathcal{L}(g)}} \tag{45}$$

The first term here is an expectation of a bounded function, $f(g)$, so is bounded, while second term converges to zero in the limit (by the previous result).

Finally, we consider the first ratio in the numerator,

$$\frac{\int_{A(\Delta)} dg \, f(g)h(g)e^{N\mathcal{L}(g)}}{\int_{A(\Delta)} dg \, h(g)e^{N\mathcal{L}(g)}} \tag{46}$$

which can be understood as an expectation over $f(g)$ in the region $A(\Delta)$. As $f$ is continuous, for any $\epsilon > 0$, we can find a $\delta > 0$ such that for all $g$ with $|g^* - g| < \delta$, we have

$$f(g^*) - \epsilon < f(g) < f(g^*) + \epsilon. \tag{47}$$

Further, because the continuous function, $\mathcal{L}(g)$, has a unique global optimum, $g^*$, for every $\delta > 0$ we are always able to find a $\Delta > 0$ such that all points $g \in A(\Delta)$ are within $\delta$ of $g^*$ i.e. $|g^* - g| < \delta$. Thus combining the previous two facts, given an $\epsilon$, we are always able to find a $\delta$ such that Eq. 47 holds for all $g$ with $|g^* - g| < \delta$, and given a $\delta$ we are always able to find a $\Delta$ such that all $g \in A(\Delta)$ have $|g^* - g| < \delta$. Hence for every $\epsilon > 0$ we can find a $\Delta > 0$ such that Eq. 47 holds for all $g \in A(\Delta)$. Choosing the appropriate $\epsilon$-dependent $\Delta$ and substituting Eq. 47 into Eq. 46, $\epsilon$ also bounds the error in the expectation,

$$f(g^*) - \epsilon < \frac{\int_{A(\Delta)} dg \; f(g) h(g) e^{N\mathcal{L}(g)}}{\int_{A(\Delta)} dg \; h(g) e^{N\mathcal{L}(g)}} < f(g^*) + \epsilon. \tag{48}$$

Now, we use the results in Eq. (44), Eq. (45) and Eq. (48) to take the limit of Eq. (43) (we can compose these limits by the algebraic limit theorem as all the individual limits exist and are finite),

$$f(g^*) - \epsilon < \lim_{N \to \infty} \mathbb{E}\left[f(g)\right] < f(g^*) + \epsilon. \tag{49}$$

And as this holds for any $\epsilon$, we have,

$$f(g^*) = \lim_{N \to \infty} \mathbb{E}\left[f(g)\right]. \tag{50}$$

This result is applicable to the DGP posterior over Gram matrices, as that posterior can be written as,

$$\mathrm{P}_N\left(\mathbf{G}_1, \dots, \mathbf{G}_L | \mathbf{X}, \tilde{\mathbf{Y}}\right) \propto h(g) e^{N\mathcal{L}(g)}, \tag{51}$$

where $\mathcal{L}(g)$ is the usual DKM objective,

$$\mathcal{L}(g) = \mathcal{L}(\mathbf{G}_1, \dots, \mathbf{G}_L) \tag{52}$$

and $h(g)$ is the remaining terms in the log-posterior which do not depend on $N$,

$$h(g) = \exp\left(-\tfrac{P+1}{2} \sum_{\ell} \log |\mathbf{G}_\ell|\right) \tag{53}$$

(this requires $P \leq N$ so that $\mathbf{G}_\ell$ is full-rank).

## D  GENERAL MODELS IN THE BAYESIAN REPRESENTATION LEARNING LIMIT

Overall, our goal is to compute the integral in Eq. (6) in the limit as $N \to \infty$. While the integral is intractable for general models such as BNNs, we can use variational inference to reason about its properties. In particular, we can bound the integral using the ELBO,

$$\log \mathrm{P}\left(\mathbf{G}_\ell | \mathbf{G}_{\ell-1}\right) \geq \mathrm{ELBO}_\ell = \mathbb{E}_{\mathrm{Q}(\mathbf{F}_\ell)}\left[\log \mathrm{P}\left(\mathbf{G}_\ell | \mathbf{F}_\ell\right) + \log \mathrm{P}\left(\mathbf{F}_\ell | \mathbf{G}_{\ell-1}\right) - \log \mathrm{Q}\left(\mathbf{F}_\ell\right)\right]. \tag{54}$$

Note that $\mathrm{Q}\left(\mathbf{F}_\ell\right)$ here is different from $\mathrm{Q}_{\theta_\ell}\left(\mathbf{f}_\lambda^\ell\right)$ in the main text, both because the approximate posterior here, $\mathrm{Q}\left(\mathbf{F}_\ell\right)$ is over all features jointly, $\mathbf{F}_\ell$, whereas the approximate posterior in the main text is only over a single feature, $\mathbf{f}_\lambda^\ell$, and because in the main text, we chose a specific family of distribution with parameters $\theta_\ell$, while here we leave the approximate posterior, $\mathrm{Q}\left(\mathbf{F}_\ell\right)$ completely unconstrained, so that it has the flexibility to capture the true posterior. Indeed, if the optimal approximate posterior is equal to the true posterior, $\mathrm{Q}^*\left(\mathbf{F}_\ell\right) = \mathrm{P}\left(\mathbf{F}_\ell | \mathbf{G}_{\ell-1}, \mathbf{G}_\ell\right)$, then the bound is tight, so we get $\log \mathrm{P}\left(\mathbf{G}_\ell | \mathbf{G}_{\ell-1}\right) = \mathrm{ELBO}_\ell^*$. Our overall strategy is thus to use variational inference to characterise the optimal approximate which is equal to the true posterior $\mathrm{Q}^*\left(\mathbf{F}_\ell\right) = \mathrm{P}\left(\mathbf{F}_\ell | \mathbf{G}_{\ell-1}, \mathbf{G}_\ell\right)$ and use the corresponding ELBO to obtain $\log \mathrm{P}\left(\mathbf{G}_\ell | \mathbf{G}_{\ell-1}\right)$.

### D.1  CHARACTERISING EXACT BNN POSTERIORS

Remember that if the approximate posterior family, $\mathrm{Q}\left(\mathbf{F}_\ell\right)$ is flexible enough to capture the true posterior $\mathrm{P}\left(\mathbf{F}_\ell | \mathbf{G}_{\ell-1}, \mathbf{G}_\ell\right)$, then the $\mathrm{Q}^*\left(\mathbf{F}_\ell\right)$ that optimizes the ELBO is indeed the true posterior, the bound is tight, so the ELBO is equal to $\log \mathrm{P}\left(\mathbf{G}_\ell | \mathbf{G}_{\ell-1}\right)$ (Jordan et al., 1999; Blei et al., 2017). Thus, we are careful to ensure that our approximate posterior family captures the true posterior, by ensuring that we only impose constraints on $\mathrm{Q}\left(\mathbf{F}_\ell\right)$ that must hold for the true posterior,

$P(\mathbf{F}_\ell | \mathbf{G}_{\ell-1}, \mathbf{G}_\ell)$. In particular, note that $P(\mathbf{G}_\ell | \mathbf{F}_\ell)$ in Eq. (5b) constrains the true posterior to give non-zero mass only to $\mathbf{F}_\ell$ that satisfy $\mathbf{G}_\ell = \frac{1}{N_\ell}\phi(\mathbf{F}_\ell)\phi^T(\mathbf{F}_\ell)$. However, this constraint is difficult to handle. We therefore consider an alternative, weaker constraint on expectations, which holds for the true posterior (the first equality below) because Eq. (5b) constrains $\mathbf{G}_\ell = \frac{1}{N_\ell}\phi(\mathbf{F}_\ell)\phi^T(\mathbf{F}_\ell)$, and impose the same constraint on the approximate posterior,

$$\mathbf{G}_\ell = \mathbb{E}_{P(\mathbf{F}_\ell | \mathbf{G}_\ell, \mathbf{G}_{\ell-1})}\left[\frac{1}{N_\ell}\phi(\mathbf{F}_\ell)\phi^T(\mathbf{F}_\ell)\right] = \mathbb{E}_{Q(\mathbf{F}_\ell)}\left[\frac{1}{N_\ell}\phi(\mathbf{F}_\ell)\phi^T(\mathbf{F}_\ell)\right]. \tag{55}$$

Now, we can solve for the optimal $Q(\mathbf{F}_\ell)$ with this constraint on the expectation. In particular, the Lagrangian is obtained by taking the ELBO (Eq. 54), dropping the $\log P(\mathbf{G}_\ell | \mathbf{F}_\ell)$ term representing the equality constraint (that $\mathbf{G}_\ell = \frac{1}{N_\ell}\phi(\mathbf{F}_\ell)\phi^T(\mathbf{F}_\ell)$) and including Lagrange multipliers for the expectation constraint, $\mathbf{\Lambda}$, (Eq. 55) and the constraint that the distribution must normalize to 1, $\Lambda$,

$$L = \int d\mathbf{F}_\ell\, Q(\mathbf{F}_\ell)\left(\log P(\mathbf{F}_\ell | \mathbf{G}_{\ell-1}) - \log Q(\mathbf{F}_\ell)\right)$$
$$+ \tfrac{1}{2}\operatorname{Tr}\left(\mathbf{\Lambda}\left(\mathbf{G}_\ell - \int d\mathbf{F}_\ell\, Q(\mathbf{F}_\ell)\,\phi(\mathbf{F}_\ell)\phi^T(\mathbf{F}_\ell)\right)\right) + \Lambda\left(1 - \int d\mathbf{F}_\ell\, Q(\mathbf{F}_\ell)\right) \tag{56}$$

Differentiating wrt $Q(\mathbf{F}_\ell)$, and solving for the optimal approximate posterior, $Q^*(\mathbf{F}_\ell)$,

$$0 = \left.\frac{\partial L}{\partial Q(\mathbf{F}_\ell)}\right|_{Q^*(\mathbf{F}_\ell)} \tag{57}$$

$$0 = (\log P(\mathbf{F}_\ell | \mathbf{G}_{\ell-1}) - \log Q^*(\mathbf{F}_\ell)) - 1 - \tfrac{1}{2}\operatorname{Tr}\left(\mathbf{\Lambda}\phi(\mathbf{F}_\ell)\phi^T(\mathbf{F}_\ell)\right) - \Lambda \tag{58}$$

Solving for $\log Q^*(\mathbf{F}_\ell)$,

$$\log Q^*(\mathbf{F}_\ell) = \log P(\mathbf{F}_\ell | \mathbf{G}_{\ell-1}) - \tfrac{1}{2}\operatorname{Tr}\left(\mathbf{\Lambda}\phi(\mathbf{F}_\ell)\phi^T(\mathbf{F}_\ell)\right) + \text{const}. \tag{59}$$

Using the cyclic property of the trace,

$$\log Q^*(\mathbf{F}_\ell) = \log P(\mathbf{F}_\ell | \mathbf{G}_{\ell-1}) - \tfrac{1}{2}\operatorname{Tr}\left(\phi^T(\mathbf{F}_\ell)\mathbf{\Lambda}\phi(\mathbf{F}_\ell)\right) + \text{const}. \tag{60}$$

Thus, $\log Q(\mathbf{F}_\ell)$ can be written as a sum over features,

$$\log Q^*(\mathbf{F}_\ell) = \sum_{\lambda=1}^{N_\ell}\left[\log P(\mathbf{f}_\lambda^\ell | \mathbf{G}_{\ell-1}) - \tfrac{1}{2}\phi^T(\mathbf{f}_\lambda^\ell)\mathbf{\Lambda}\phi(\mathbf{f}_\lambda^\ell)\right] + \text{const} = \sum_{\lambda=1}^{N_L}\log Q(\mathbf{f}_\lambda^\ell) \tag{61}$$

so, the optimal approximate posterior is IID over features,

$$Q^*(\mathbf{F}_\ell) = \prod_{\lambda=1}^{N_\ell} Q^*(\mathbf{f}_\lambda^\ell). \tag{62}$$

Remember that this approximate posterior was only constrained in expectation, and that this constraint held for the true posterior (Eq. 55). Thus, we might think that this optimal approximate posterior would be equal to the true posterior. However, remember that the true posterior had a tighter equality constraint, that $\mathbf{G}_\ell = \frac{1}{N_\ell}\phi(\mathbf{F}_\ell)\phi^T(\mathbf{F}_\ell)$, while so far we have only imposed a weaker constraint in expectation (Eq. 55). We thus need to check that our optimal approximate posterior does indeed satisfy the equality constraint in the limit as $N_\ell \to \infty$. This be shown using the law of large numbers, as $\mathbf{f}_\lambda^\ell$ are IID under the optimal approximate posterior, and by using Eq. (55) for the final equality,

$$\lim_{N_\ell \to \infty}\frac{1}{N_\ell}\phi(\mathbf{F}_\ell)\phi^T(\mathbf{F}_\ell) = \lim_{N_\ell \to \infty}\frac{1}{N_\ell}\sum_{\lambda=1}^{N_\ell}\phi(\mathbf{f}_\lambda^\ell)\phi^T(\mathbf{f}_\lambda^\ell) = \mathbb{E}_{Q(\mathbf{f}_\lambda^\ell)}\left[\phi(\mathbf{f}_\lambda^\ell)\phi^T(\mathbf{f}_\lambda^\ell)\right] = \mathbf{G}_\ell. \tag{63}$$

Thus, the optimal approximate posterior does meet the constraint in the limit as $N_\ell \to \infty$, so in that limit, the true posterior, like the optimal approximate posterior is IID across features,

$$P(\mathbf{F}_\ell | \mathbf{G}_{\ell-1}, \mathbf{G}_\ell) = Q^*(\mathbf{F}_\ell) = \prod_{\lambda=1}^{N_\ell} Q^*(\mathbf{f}_\lambda^\ell) = \prod_{\ell=1}^{N_\ell} P(\mathbf{f}_\lambda^\ell | \mathbf{G}_{\ell-1}, \mathbf{G}_\ell). \tag{64}$$

## D.2 EXACTLY MULTIVARIATE GAUSSIAN DGP POSTERIORS

For DGPs, we have $\phi(\mathbf{f}_\lambda^\ell) = \mathbf{f}_\lambda^\ell$, so the optimal approximate posterior is Gaussian,

$$\log Q_{\text{DGP}}^* \left(\mathbf{f}_\lambda^\ell\right) = \log P_{\text{DGP}} \left(\mathbf{f}_\lambda^\ell | \mathbf{G}_{\ell-1}\right) - \tfrac{1}{2}(\mathbf{f}_\lambda^\ell)^T \mathbf{\Lambda} \mathbf{f}_\lambda^\ell + \text{const} \tag{65}$$

$$= -\tfrac{1}{2}(\mathbf{f}_\lambda^\ell)^T \left(\mathbf{\Lambda} + \mathbf{K}^{-1}(\mathbf{G}_{\ell-1})\right) \mathbf{f}_\lambda^\ell + \text{const} \tag{66}$$

$$= \log \mathcal{N} \left(\mathbf{f}_\lambda^\ell; \mathbf{0}, \left(\mathbf{\Lambda} + \mathbf{K}^{-1}(\mathbf{G}_{\ell-1})\right)^{-1}\right). \tag{67}$$

As the approximate posterior and true posterior are IID, the constraint in Eq. (55) becomes,

$$\mathbf{G}_\ell = \mathbb{E}_{P_{\text{DGP}}\left(\mathbf{f}_\lambda^\ell | \mathbf{G}_\ell, \mathbf{G}_{\ell-1}\right)} \left[\mathbf{f}_\lambda^\ell (\mathbf{f}_\lambda^\ell)^T\right] = \mathbb{E}_{Q_{\text{DGP}}^*\left(\mathbf{f}_\lambda^\ell\right)} \left[\mathbf{f}_\lambda^\ell (\mathbf{f}_\lambda^\ell)^T\right] = \left(\mathbf{\Lambda} + \mathbf{K}^{-1}(\mathbf{G}_{\ell-1})\right)^{-1}. \tag{68}$$

As the Lagrange multipliers are unconstrained, we can always set them such that this constraint holds. In that case both the optimal approximate posterior and the true posterior become,

$$P_{\text{DGP}} \left(\mathbf{f}_\lambda^\ell | \mathbf{G}_{\ell-1} \mathbf{G}_\ell\right) = Q_{\text{DGP}}^* \left(\mathbf{f}_\lambda^\ell\right) = \mathcal{N} \left(\mathbf{f}_\lambda^\ell; \mathbf{0}, \mathbf{G}_\ell\right), \tag{69}$$

as required.

## D.3 GENERAL FORM FOR THE CONDITIONAL DISTRIBUTION OVER GRAM MATRICES

Now that we have shown that the true posterior, $P\left(\mathbf{F}_\ell | \mathbf{G}_{\ell-1}, \mathbf{G}_\ell\right)$ factorises, we can obtain a simple form for $\log P\left(\mathbf{G}_\ell | \mathbf{G}_{\ell-1}\right)$. In particular, $\log P\left(\mathbf{G}_\ell | \mathbf{G}_{\ell-1}\right)$ is equal to the ELBO if we use the true posterior in place of the approximate posterior,

$$\lim_{N_\ell \to \infty} \tfrac{1}{N} \log P\left(\mathbf{G}_\ell | \mathbf{G}_{\ell-1}\right) = \lim_{N_\ell \to \infty} \tfrac{1}{N} \mathbb{E}_{P(\mathbf{F}_\ell | \mathbf{G}_{\ell-1}, \mathbf{G}_\ell)} \left[\log P\left(\mathbf{G}_\ell | \mathbf{F}_\ell\right) + \log \frac{P\left(\mathbf{F}_\ell | \mathbf{G}_{\ell-1}\right)}{P\left(\mathbf{F}_\ell | \mathbf{G}_{\ell-1}, \mathbf{G}_\ell\right)}\right]. \tag{70}$$

Under the posterior, the constraint represented by $\log P\left(\mathbf{G}_\ell | \mathbf{F}_\ell\right)$ is satisfied, so in the limit we can include that term in a constant,

$$\lim_{N_\ell \to \infty} \tfrac{1}{N} \log P\left(\mathbf{G}_\ell | \mathbf{G}_{\ell-1}\right) = \lim_{N_\ell \to \infty} \tfrac{1}{N} \mathbb{E}_{P(\mathbf{F}_\ell | \mathbf{G}_{\ell-1}, \mathbf{G}_\ell)} \left[\log \frac{P\left(\mathbf{F}_\ell | \mathbf{G}_{\ell-1}\right)}{P\left(\mathbf{F}_\ell | \mathbf{G}_{\ell-1}, \mathbf{G}_\ell\right)}\right] + \text{const}. \tag{71}$$

Now, we use the fact that the prior, $P\left(\mathbf{F}_\ell | \mathbf{G}_{\ell-1}\right)$ and posterior, $P\left(\mathbf{F}_\ell | \mathbf{G}_{\ell-1}, \mathbf{G}_\ell\right)$, are IID across features,

$$\lim_{N_\ell \to \infty} \tfrac{1}{N} \log P\left(\mathbf{G}_\ell | \mathbf{G}_{\ell-1}\right) = \nu_\ell \mathbb{E}_{P\left(\mathbf{f}_\lambda^\ell | \mathbf{G}_{\ell-1}, \mathbf{G}_\ell\right)} \left[\log \frac{P\left(\mathbf{f}_\lambda^\ell | \mathbf{G}_{\ell-1}\right)}{P\left(\mathbf{f}_\lambda^\ell | \mathbf{G}_{\ell-1}, \mathbf{G}_\ell\right)}\right] + \text{const} \tag{72}$$

and this expectation is a KL-divergence,

$$\lim_{N_\ell \to \infty} \tfrac{1}{N} \log P\left(\mathbf{G}_\ell | \mathbf{G}_{\ell-1}\right) = -\nu_\ell \, D_{\text{KL}} \left(P\left(\mathbf{f}_\lambda^\ell | \mathbf{G}_{\ell-1}, \mathbf{G}_\ell\right) \| P\left(\mathbf{f}_\lambda^\ell | \mathbf{G}_{\ell-1}\right)\right) + \text{const}, \tag{73}$$

which gives Eq. (32) when we combine with Eq. (8).

## D.4 PARAMETRIC APPROXIMATE POSTERIORS

Eq. (64) represents a considerable simplification, as we now need to consider only a single feature, $\mathbf{f}_\lambda^\ell$, rather than the joint distribution over all features, $\mathbf{F}_\ell$. However, in the general case, it is still not possible to compute Eq. (64) because the true posterior over a single feature is still not tractable. Following the true posteriors derived in the previous section, we could chose a parametric approximate posterior that factorises across features,

$$Q_\theta \left(\mathbf{F}_1, \ldots, \mathbf{F}_L\right) = \prod_{\ell=1}^{L} \prod_{\lambda=1}^{N_\ell} Q_{\theta_\ell} \left(\mathbf{f}_\lambda^\ell\right). \tag{74}$$

Remember that we optimize the approximate posterior parameters, $\theta$, directly, and set the Gram matrices as a function of $\theta$ (Eq. 34). As before, we can bound, $\log P\left(\mathbf{G}_\ell = \mathbf{G}_\theta(\theta_\ell) | \mathbf{G}_{\ell-1}\right)$ using the

ELBO, and the bound is tight when the approximate posterior equals the true posterior,

$$\log P\left(\mathbf{G}_\ell = \mathbf{G}_\theta(\theta_\ell)|\mathbf{G}_{\ell-1}\right) \tag{75}$$

$$= \mathbb{E}_{P\left(\mathbf{F}_\ell|\mathbf{G}_{\ell-1},\mathbf{G}_\ell=\mathbf{G}_\theta(\theta_\theta)\right)}\left[\log P\left(\mathbf{G}_\ell=\mathbf{G}_\theta(\theta_\ell)|\mathbf{F}_\ell\right) + \log \frac{P\left(\mathbf{F}_\lambda^\ell|\mathbf{G}_{\ell-1}\right)}{P\left(\mathbf{F}_\ell|\mathbf{G}_{\ell-1},\mathbf{G}_\ell=\mathbf{G}_\theta(\theta_\ell)\right)}\right] \tag{76}$$

$$\geq \mathbb{E}_{Q_\theta(\mathbf{F}_\ell)}\left[\log P\left(\mathbf{G}_\ell=\mathbf{G}_\theta(\theta_\ell)|\mathbf{F}_\ell\right) + \log \frac{P\left(\mathbf{F}_\lambda^\ell|\mathbf{G}_{\ell-1}\right)}{Q_{\theta_\ell}\left(\mathbf{F}_\ell\right)}\right]. \tag{77}$$

Now, we can cancel the $\log P\left(\mathbf{G}_\ell = \mathbf{G}_\theta(\theta_\ell)|\mathbf{F}_\ell\right)$ terms, as they represent a constraint that holds both under the true posterior, and under the approximate posterior,

$$\mathbb{E}_{P\left(\mathbf{F}_\ell|\mathbf{G}_{\ell-1},\mathbf{G}_\ell=\mathbf{G}_\theta(\theta_\ell))\right)}\left[\log \frac{P\left(\mathbf{F}_\ell|\mathbf{G}_{\ell-1}\right)}{P\left(\mathbf{F}_\ell|\mathbf{G}_{\ell-1},\mathbf{G}_\ell=\mathbf{G}_\theta(\theta_\ell)\right)}\right] \geq \mathbb{E}_{Q_{\theta_\ell}(\mathbf{F}_\ell)}\left[\log \frac{P\left(\mathbf{F}_\ell|\mathbf{G}_{\ell-1}\right)}{Q_{\theta_\ell}\left(\mathbf{F}_\ell\right)}\right]. \tag{78}$$

Using the fact that the prior, posterior and approximate posterior are all IID over features, we can write this inequality in terms of distributions over a single feature, $\mathbf{f}_\lambda^\ell$ and divide by $N_\ell$,

$$\mathbb{E}_{P\left(\mathbf{f}_\lambda^\ell|\mathbf{G}_{\ell-1},\mathbf{G}_\ell=\mathbf{G}_\theta(\theta_\ell)\right)}\left[\log \frac{P\left(\mathbf{f}_\lambda^\ell|\mathbf{G}_{\ell-1}\right)}{P\left(\mathbf{f}_\lambda^\ell|\mathbf{G}_{\ell-1},\mathbf{G}_\ell=\mathbf{G}_\theta(\theta_\ell)\right)}\right] \geq \mathbb{E}_{Q_{\theta_\ell}(\mathbf{f}_\lambda^\ell)}\left[\log \frac{P\left(\mathbf{f}_\lambda^\ell|\mathbf{G}_{\ell-1}(\theta)\right)}{Q_{\theta_\ell}\left(\mathbf{f}_\lambda^\ell\right)}\right]. \tag{79}$$

Noting that both sides of this inequality are negative KL-divergences, we obtain,

$$- D_{KL}\left(P\left(\mathbf{f}_\lambda^\ell|\mathbf{G}_{\ell-1},\mathbf{G}_\ell=\mathbf{G}_\theta(\theta_\ell)\right)\big\|P\left(\mathbf{f}_\lambda^\ell|\mathbf{G}_{\ell-1}\right)\right) \geq - D_{KL}\left(Q_{\theta_\ell}\left(\mathbf{f}_\lambda^\ell\right)\big\|P\left(\mathbf{f}_\lambda^\ell|\mathbf{G}_{\ell-1}\right)\right), \tag{80}$$

which gives Eq. (33) in the main text.

# E THEORETICAL SIMILARITIES IN REPRESENTATION LEARNING IN FINITE AND INFINITE NETWORKS

In the main text, we considered probability densities of the Gram matrices, $\mathbf{G}_1, \ldots, \mathbf{G}_L$. However, we can also consider probability densities of the features, $\mathbf{F}_1, \ldots, \mathbf{F}_L$, for a DGP,

$$\log P\left(\mathbf{F}_\ell|\mathbf{F}_{\ell-1}\right) = -\tfrac{N_\ell}{2}\log|\mathbf{K}\left(\mathbf{G}_{\text{DGP}}\left(\mathbf{F}_{\ell-1}\right)\right)| - \tfrac{1}{2}\operatorname{tr}\left(\mathbf{F}_\ell^T\mathbf{K}^{-1}\left(\mathbf{G}_{\text{DGP}}\left(\mathbf{F}_{\ell-1}\right)\right)\mathbf{F}_\ell\right) + \text{const}. \tag{81}$$

We can rewrite the density such that it is still the density of features, $\mathbf{F}_\ell$, but it is expressed in terms of the DGP Gram matrix,

$$\log P\left(\mathbf{F}_\ell|\mathbf{F}_{\ell-1}\right) = -\tfrac{N_\ell}{2}\log|\mathbf{K}(\mathbf{G}_{\ell-1})| - \tfrac{N_\ell}{2}\operatorname{tr}\left(\mathbf{K}^{-1}(\mathbf{G}_{\ell-1})\mathbf{G}_\ell\right) + \text{const}. \tag{82}$$

Here, we have used the cyclic property of the trace to combine the $\mathbf{F}_\ell$ and $\mathbf{F}_\ell^T$ to form $\mathbf{G}_\ell$, and we have used the fact that our kernels can be written as a function of the Gram matrix. Overall, we can therefore write the posterior over features, $P\left(\mathbf{F}_1, \ldots, \mathbf{F}_L|\mathbf{X}, \tilde{\mathbf{Y}}\right)$, in terms of only Gram matrices,

$$\mathcal{J}(\mathbf{G}_1, \ldots, \mathbf{G}_L) = \tfrac{1}{N}\log P\left(\mathbf{F}_1, \ldots, \mathbf{F}_L|\mathbf{X}, \tilde{\mathbf{Y}}\right) = \log P\left(\mathbf{Y}|\mathbf{G}_L\right) + \tfrac{1}{N}\sum_{\ell=1}^{L}\log P\left(\mathbf{F}_\ell|\mathbf{F}_{\ell-1}\right), \tag{83}$$

substituting Eq. (82),

$$\mathcal{J}(\mathbf{G}_1, \ldots, \mathbf{G}_L) = \log P\left(\mathbf{Y}|\mathbf{G}_L\right) - \tfrac{1}{2}\sum_{\ell=1}^{L}\nu_\ell\left(\log|\mathbf{K}(\mathbf{G}_{\ell-1})| + \operatorname{tr}\left(\mathbf{K}^{-1}(\mathbf{G}_{\ell-1})\mathbf{G}_\ell\right)\right) + \text{const}. \tag{84}$$

Thus, $\mathcal{J}(\mathbf{G}_1, \ldots, \mathbf{G}_L)$ does not depend on $N$, and thus the Gram matrices that maximize $\mathcal{J}(\mathbf{G}_1, \ldots, \mathbf{G}_L)$ are the same for any choice of $N$. The only restriction is that we need $N_\ell \geq P$, to ensure that the Gram matrices are full-rank.

To confirm these results, we used Adam with a learning rate of $10^{-3}$ to optimize full-rank Gram matrices with Eq. (84) and to directly do MAP inference over features using Eq. (81). As expected, as the number of features increased, the Gram matrix from MAP inference over features converged rapidly to that expected using Eq. (84) (Fig. 5).

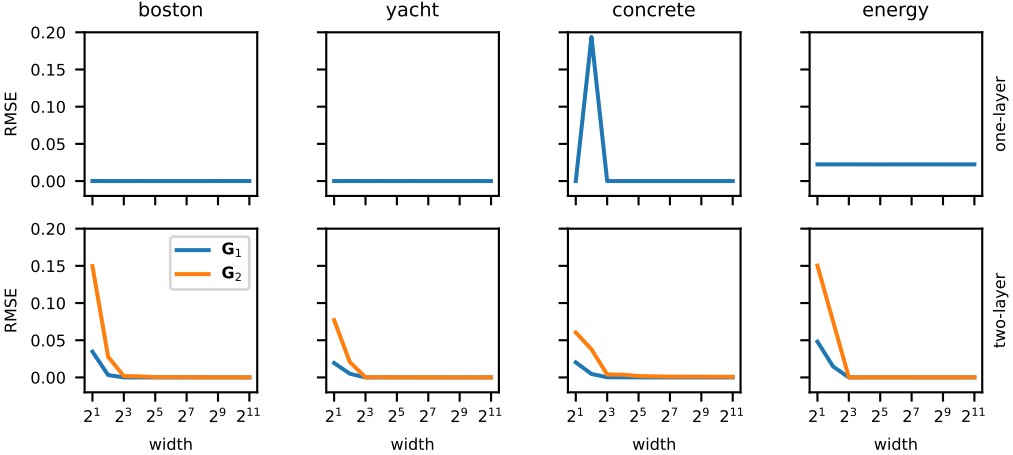

Figure 5: RMSE of trained Gram matrices between one-hidden-layer (first row) and two-hidden-layer (second row) DGPs of various width trained by gradient descent and the corresponding MAP limit. Columns correspond to different datasets (trained on a subset of 50 datapoints).

## F    ADDITIONAL EXPERIMENTAL DETAILS

To optimize the analytic DKM objective for DGPs and the variational DKM objective for DGPs (Figs. 3–11), we parameterised the Gram matrices (or covariances for the variational approximate posterior) as the product of a square matrix, $\mathbf{R}_\ell \in \mathbb{R}^{P \times P}$, with itself transposed, $\mathbf{G}_\ell = \frac{1}{P} \mathbf{R}_\ell \mathbf{R}_\ell^T$, and we used Adam with a learning rate of $10^{-3}$ to learn $\mathbf{R}_\ell$. To do Bayesian inference in finite BNNs and DGPs, we used Langevin sampling with 10 parallel chains, and a step size of $10^{-3}$. Note that in certain senarios, Langevin sampling can be very slow, as the features have a Gaussian prior with covariance $\mathbf{K}(\mathbf{G}_{\ell-1})$ which has some very small and some larger eigenvalues, which makes sampling difficult. Instead, we reparameterised the model in terms of the standard Gaussian random variables, $\mathbf{V}_\ell \in \mathbb{R}^{P \times N_\ell}$. We then wrote $\mathbf{F}_\ell$ in terms of $\mathbf{V}_\ell$,

$$\mathbf{F}_\ell = \mathbf{L}_{\ell-1} \mathbf{V}_\ell. \tag{85}$$

Here, $\mathbf{L}_{\ell-1}$ is the Cholesky of $\mathbf{K}(\mathbf{G}_{\ell-1})$, so $\mathbf{K}(\mathbf{G}_{\ell-1}) = \mathbf{L}_{\ell-1} \mathbf{L}_{\ell-1}^T$. This gives an equivalent distribution $\mathrm{P}\left(\mathbf{F}_\ell | \mathbf{F}_{\ell-1}\right)$. Importantly, as the prior on $\mathbf{V}_\ell$ is IID standard Gaussian, sampling $\mathbf{V}_\ell$ is much faster. To ensure that the computational cost of these expensive simulations remained reasonable, we used a subset of 50 datapoints from each dataset.

For the DKM objective for BNNs, we used Monte-Carlo to approximate the Gram matrices,

$$\mathbf{G}_\theta(\theta_\ell) \approx \sum_{k=1}^{K} \phi(\mathbf{f}_k^\ell) \phi^T(\mathbf{f}_k^\ell). \tag{86}$$

with $\mathbf{f}_k^\ell$ drawn from the appropriate approximate posterior, and $K = 2^{16}$. We can use the reparameterisation trick (Kingma & Welling, 2013; Rezende et al., 2014) to differentiate through these Monte-Carlo estimates.

## G    ADDITIONAL COMPARISONS WITH FINITE-WIDTH DGPS

Here, we give additional results supporting those in Sec. 4.6, Fig. 3–Fig. 11. In particular, we give the DGP representations learned by two-layer networks on all UCI datasets (boston, concrete, energy, yacht), except those already given in the main text Fig. 6–8.

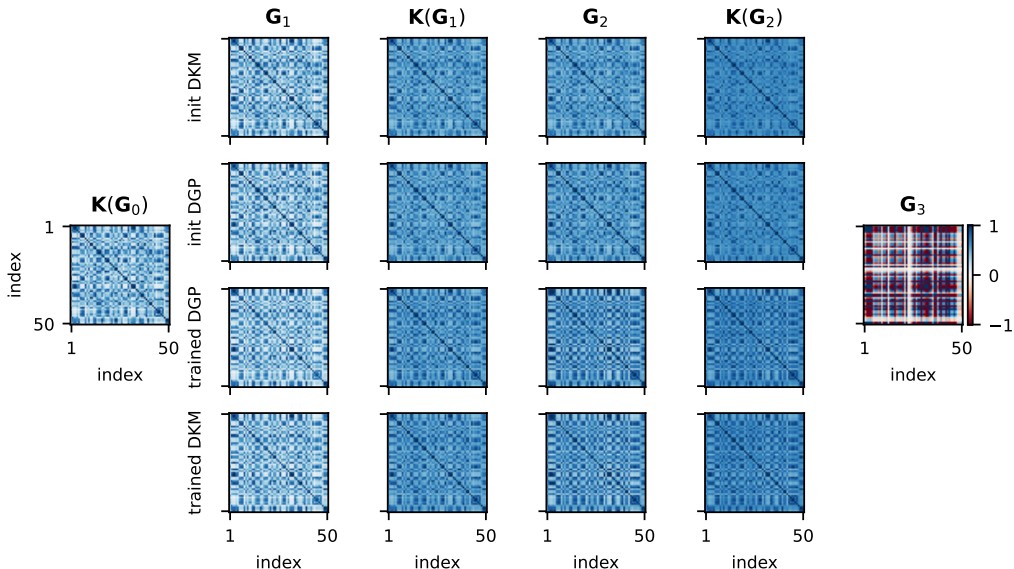

Figure 6: One hidden layer DGP and DKM with squared exponential kernel trained on a subset of energy. First and second row: initializations of DGP and DKM. Third and fourth row: trained DGP (by Langevin sampling) and DKM Gram matrices and kernels.

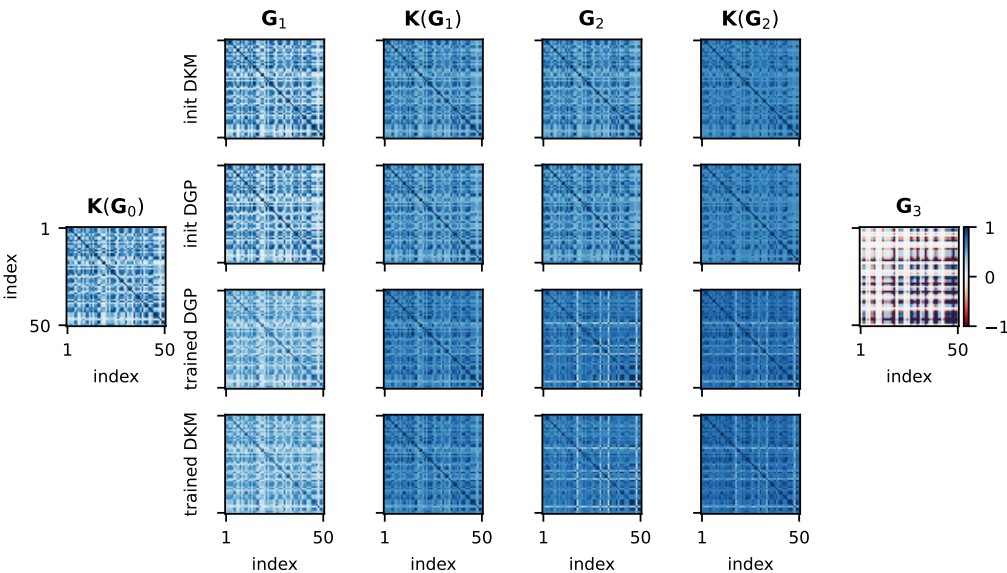

Figure 7: One hidden layer DGP and DKM with squared exponential kernel trained on a subset of boston. First and second row: initializations of DGP and DKM. Third and fourth row: trained DGP (by Langevin sampling) and DKM Gram matrices and kernels.

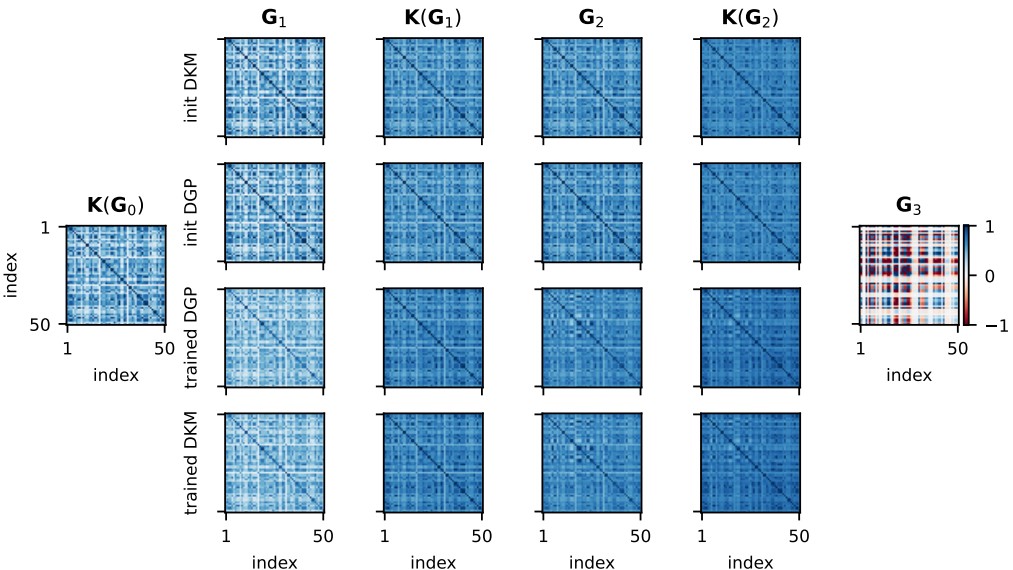

Figure 8: One hidden layer DGP and DKM with squared exponential kernel trained on a subset of concrete. First and second row: initializations of DGP and DKM. Third and fourth row: trained DGP (by Langevin sampling) and DKM Gram matrices and kernels.

# H  THE FLOW POSTERIOR IN A 2-LAYER BNN

Here, we give the 2-layer version (Fig. 9) of Fig. 4 in the main text, which again shows a close match between the variational DKM with a flow posterior, and the BNN true posterior.

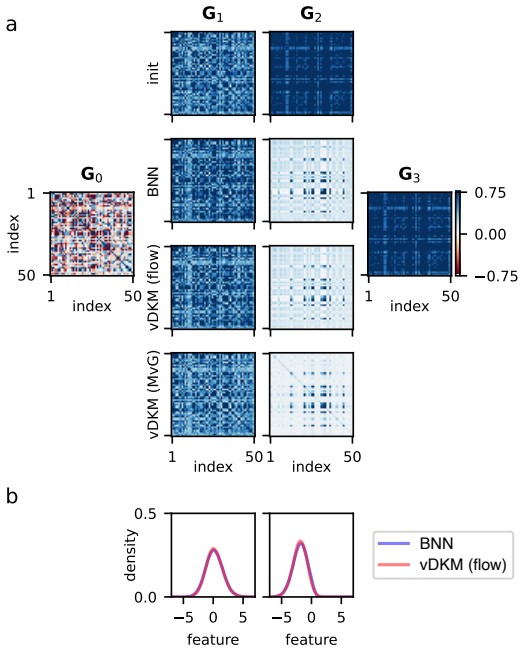

Figure 9: Two-layer ReLU BNN and variational DKM with flow. **a** Initialized (first row) and learned Gram matrices of a width 1024 BNN (second row), vDKM with flow (third row) and vDKM with multivariate Gaussian (fourth row) using $2^{14}$ Monte-Carlo samples. The Gram matrices between BNN and vDKM (flow) match closely after training. (MvG). **b** Marginal PDF over features at each layer for one input datapoint using kernel density estimation. The marginal PDFs of BNN are non-Gaussian (blue curves), vDKM with flow is able to capture the non-Gaussianity and match closely with BNNs marginals (red curves).

# I MULTIVARIATE GAUSSIAN APPROXIMATE POSTERIORS IN DEEPER NETWORKS

There is a body of theoretical work (e.g. (Seroussi & Ringel, 2021)), on BNNs that approximates BNN posteriors over features as Gaussian. While we have shown that this is a bad idea in general (Fig. 4 and 9), we can nonetheless ask whether there are circumstances where the idea might work well. In particular, we hypothesised that depth is an important factor. In particular, in shallow networks, in order to get $\mathbf{G}_L$ close to the required representation, we may need the posterior over $\mathbf{F}_\ell$ to be quite different from the prior. In contrast, in deeper networks, we might expect the posterior over $\mathbf{F}_\ell$ to be closer to its (Gaussian) prior, and therefore we might Gaussian approximate posteriors to work better.

However, we cannot just make the network deeper, because as we do so, we apply the nonlinearity more times and dramatically alter the network's inductive biases. To resolve this issue, we derive a leaky relu nonlinearity that allows (approximately) independent control over the inductive biases (or effective depth) and the actual depth (Appendix I.1). Using these nonlinearities, we indeed find that very deep networks are reasonably well approximated by multivariate Gaussian approximate posteriors (Appendix I.2).

## I.1 LEAKY RELU NONLINEARITIES

Our goal is to find a pointwise nonlinearity, $\phi$, such that (under the prior),

$$\mathbb{E}_{\mathrm{P}_{\mathrm{DGP}}\left(\mathbf{f}_\lambda^\ell | \mathbf{G}_{\ell-1}\right)}\left[\phi(\mathbf{f}_\lambda^\ell)\phi^T(\mathbf{f}_\lambda^\ell)\right] = p\,\mathbb{E}_{\mathrm{P}\left(\mathbf{f}_\lambda^\ell | \mathbf{G}_{\ell-1}\right)}\left[\mathrm{relu}(\mathbf{f}_\lambda^\ell)\mathrm{relu}^T(\mathbf{f}_\lambda^\ell)\right] + (1-p)\mathbf{G}_{\ell-1}. \tag{87}$$

We will set $p = \alpha/L$, where $\alpha$ is the "effective" depth of the network and $L$ is the real depth. These networks are designed such that their inductive biases in the infinite width limit are similar to a standard relu network with depth $\alpha$. Indeed, we would take this approach if we wanted a well-defined infinite-depth DKM limit.

Without loss of generality, we consider a 2D case, where $x$ and $y$ are zero-mean bivariate Gaussian,

$$\pi(x, y) = \mathcal{N}\left(\begin{pmatrix} x \\ y \end{pmatrix}; \mathbf{0}, \begin{pmatrix} \Sigma_{\mathrm{xx}} & \Sigma_{\mathrm{xy}} \\ \Sigma_{\mathrm{xy}} & \Sigma_{\mathrm{yy}} \end{pmatrix}\right) \tag{88}$$

where $\pi(x, y)$ is the probability density for the joint distribution. Note that we use a scaled relu,

$$\mathrm{relu}(x) = \begin{cases} \sqrt{2}\,x & \text{for } 0 < x \\ 0 & \text{otherwise} \end{cases} \tag{89}$$

such that $\mathbb{E}\left[\mathrm{relu}^2(x)\right] = \Sigma_{\mathrm{xx}}$. Mirroring Eq. 87, we want the nonlinearity, $\phi$, to satisfy,

$$\mathbb{E}\left[\phi(x^2)\right] = p\,\mathbb{E}\left[\mathrm{relu}^2(x)\right] + (1-p)\Sigma_{\mathrm{xx}} = \Sigma_{\mathrm{xx}} \tag{90a}$$

$$\mathbb{E}\left[\phi(y^2)\right] = p\,\mathbb{E}\left[\mathrm{relu}^2(y)\right] + (1-p)\Sigma_{\mathrm{yy}} = \Sigma_{\mathrm{yy}} \tag{90b}$$

$$\mathbb{E}\left[\phi(x)\phi(y)\right] = p\,\mathbb{E}\left[\mathrm{relu}(x)\mathrm{relu}(y)\right] + (1-p)\Sigma_{\mathrm{xy}} \tag{90c}$$

We hypothesise that this nonlinearity has the form,

$$\phi(x) = a\,\mathrm{relu}(x) + bx. \tag{91}$$

We will write the relu as a sum of $x$ and $|x|$,

$$\mathrm{relu}(x) = \tfrac{1}{\sqrt{2}}(x + |x|), \tag{92}$$

because $\mathbb{E}\left[f(x, y)\right] = 0$ for $f(x, y) = x|y|$ or $f(x, y) = |x|y$. It turns out that we get zero expectation for all functions where $f(-x, -y) = -f(x, y)$, which holds for the two choices above. To show such functions have a zero expectation, we write out the integral explicitly,

$$\mathbb{E}\left[f(x, y)\right] = \int_{-\infty}^{\infty} dx \int_{-\infty}^{\infty} dy\; \pi(x, y)f(x, y). \tag{93}$$

We split the domain of integration for $y$ at zero,

$$\mathbb{E}\left[f(x,y)\right] = \int_{-\infty}^{\infty} dx \int_{-\infty}^{0} dy\, \pi(x,y)f(x,y) + \int_{-\infty}^{\infty} dx \int_{0}^{\infty} dy\, \pi(x,y)f(x,y). \tag{94}$$

We substitute $y' = -y$ and $x' = -x$ in the first integral,

$$\mathbb{E}\left[f(x,y)\right] = \int_{-\infty}^{\infty} dx' \int_{0}^{\infty} dy'\, \pi(-x',-y')f(-x',-y') + \int_{-\infty}^{\infty} dx \int_{0}^{\infty} dy\, \pi(x,y)f(x,y). \tag{95}$$

As the variables we integrate over are arbitrary we can relabel $y'$ as $y$ and $x'$ as $x$, and we can then merge the integrals as their limits are the same,

$$\mathbb{E}\left[f(x,y)\right] = \int_{-\infty}^{\infty} dx \int_{0}^{\infty} dy\, \left[\pi(-x,-y)f(-x,-y) + \pi(x,y)f(x,y)\right]. \tag{96}$$

Under a zero-mean Gaussian, $\pi(-x,-y) = \pi(x,y)$,

$$\mathbb{E}\left[f(x,y)\right] = \int_{-\infty}^{\infty} dx \int_{0}^{\infty} dy\, \pi(x,y)\left(f(-x,-y) + f(x,y)\right). \tag{97}$$

Thus, if $f(-x,-y) = -f(x,y)$, then the expectation of that function under a bivariate zero-mean Gaussian distribution is zero.

Remember that our overall goal was to design a nonlinearity, $\phi$, (Eq. 91) which satisfied Eq. (90). We therefore compute the expectation,

$$\mathbb{E}\left[\phi(x)\phi(y)\right] = \mathbb{E}\left[\left(a\,\mathrm{relu}(x) + bx\right)\left(a\,\mathrm{relu}(y) + by\right)\right] \tag{98}$$

$$= \mathbb{E}\left[\left(\tfrac{a}{\sqrt{2}}\left(x + |x|\right) + bx\right)\left(\tfrac{a}{\sqrt{2}}\left(y + |y|\right) + by\right)\right] \tag{99}$$

Using the fact that $\mathbb{E}\left[\,x|y|\,\right] = \mathbb{E}\left[\,|x|y\,\right] = 0$ under a multivariate Gaussian,

$$= \mathbb{E}\left[a^2 \tfrac{1}{\sqrt{2}}\left(x + |x|\right)\tfrac{1}{\sqrt{2}}\left(y + |y|\right) + \left(\sqrt{2}ab + b^2\right)xy\right] \tag{100}$$

$$= a^2\,\mathbb{E}\left[\mathrm{relu}(x)\mathrm{relu}(y)\right] + \left(\sqrt{2}ab + b^2\right)\mathbb{E}\left[xy\right]. \tag{101}$$

Thus, we can find the value of $a$ by comparing with Eq. (90c),

$$p = a^2 \qquad\qquad\qquad a = \sqrt{p}. \tag{102}$$

For $b$, things are a bit more involved,

$$1 - p = \sqrt{2}ab + b^2 = \sqrt{2p}\,b + b^2 \tag{103}$$

where we substitute for the value of $a$. This can be rearranged to form a quadratic equation in $b$,

$$0 = b^2 + \sqrt{2p}\,b + (p - 1), \tag{104}$$

which can be solved,

$$b = \tfrac{1}{2}\left(-\sqrt{2p} \pm \sqrt{2p - 4(p-1)}\right) \tag{105}$$

$$b = \tfrac{1}{2}\left(-\sqrt{2p} \pm \sqrt{4 - 2p}\right) \tag{106}$$

$$b = -\sqrt{\tfrac{p}{2}} \pm \sqrt{1 - \tfrac{p}{2}} \tag{107}$$

Only the positive root is of interest,

$$b = \sqrt{1 - \tfrac{p}{2}} - \sqrt{\tfrac{p}{2}} \tag{108}$$

Thus, the nonlinearity is,

$$\phi(x) = \sqrt{p}\,\mathrm{relu}(x) + \left(\sqrt{1 - \tfrac{p}{2}} - \sqrt{\tfrac{p}{2}}\right)x \tag{109}$$

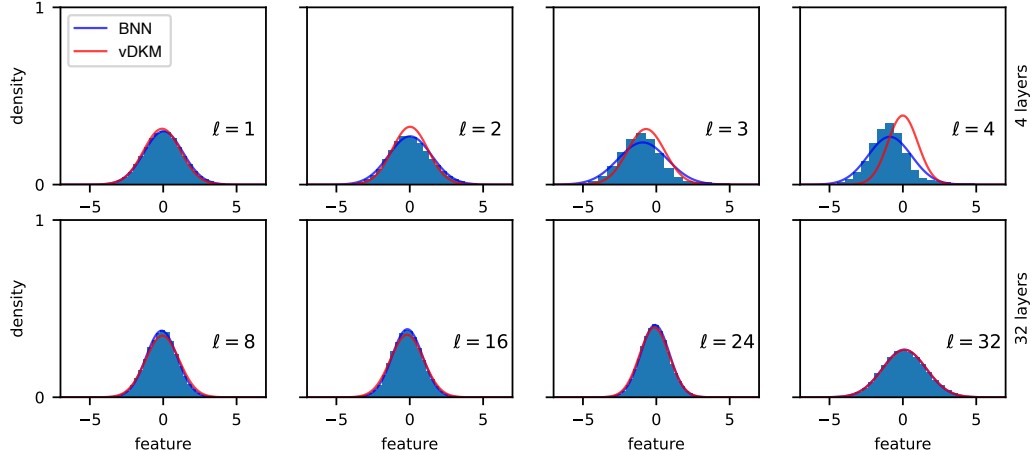

Figure 10: Comparison of posterior feature marginal distributions between a BNN of width 1024 (trained by Langevin sampling over features) and a variational DKM with $2^{16}$ Monte-Carlo samples, in a 4-layer (row 1) and a 32-layer (row 2) network. We give the BNN posterior features from Langevin sampling (blue histogarm) and the best fitting Gaussian (blue line), and compare against the variational DKM approximate posterior Gaussian distribution (red line).

where we set $p = \alpha/L$, and remember we used the scaled relu in Eq. (89). Finally, we established these choices by considering only the cross term, $\mathbb{E}\left[\phi(x)\phi(y)\right]$. We also need to check that the $\mathbb{E}\left[\phi^2(x)\right]$ and $\mathbb{E}\left[\phi^2(y)\right]$ terms are as required (Eq. 90a and Eq. 90b). In particular,

$$\mathbb{E}\left[\phi^2(x)\right] = \mathbb{E}\left[(a\,\mathrm{relu}(x) + bx)^2\right] = \mathbb{E}\left[\left(\tfrac{a}{\sqrt{2}}\left(x + |x|\right) + bx\right)^2\right] \tag{110}$$

using $\mathbb{E}\left[x|x|\right] = 0$ as $x|x|$ is an odd function of $x$, and the zero-mean Gaussian is an even distribution,

$$\mathbb{E}\left[\phi^2(x)\right] = a^2\,\mathbb{E}\left[\mathrm{relu}^2(x)\right] + \left(\sqrt{2}ab + b^2\right)\Sigma_{\mathrm{xx}} \tag{111}$$

using Eq. (102) to identify $a^2$ and Eq. (103) to identify $\sqrt{2}ab + b^2$,

$$\mathbb{E}\left[\phi^2(x)\right] = p\,\mathbb{E}\left[\mathrm{relu}^2(x)\right] + (1-p)\Sigma_{\mathrm{xx}}, \tag{112}$$

as required.

## I.2   Multivariate Gaussian in deeper networks

In the main text, we show that a more complex approximate posterior can match the distributions in these networks. Here, we consider an alternative approach. In particular, we hypothesise that these distributions are strongly non-Gaussian because the networks are shallow, meaning that the posterior needs to be far from the prior in order to get a top-layer kernel close to $\mathbf{G}_{L+1}$. We could therefore make the posteriors closer to Gaussian by using leaky-relu nonlinearities (Appendix I.1) with fixed effective depth ($\alpha = 2$), but increasing real depth, $L$. In particular, we use multivariate Gaussian approximate posteriors with learned means,

$$Q_{\theta_\ell}\left(\mathbf{f}_\lambda^\ell\right) = \mathcal{N}\left(\mathbf{f}_\lambda^\ell; \boldsymbol{\mu}_\ell, \boldsymbol{\Sigma}_\ell\right) \qquad\qquad \text{so} \qquad\qquad \theta_\ell = (\boldsymbol{\mu}_\ell, \boldsymbol{\Sigma}_\ell). \tag{113}$$

As expected, for a depth 32 network, we have much more similar marginals (Fig. 10 top) and learned representations (Fig. 11 top).

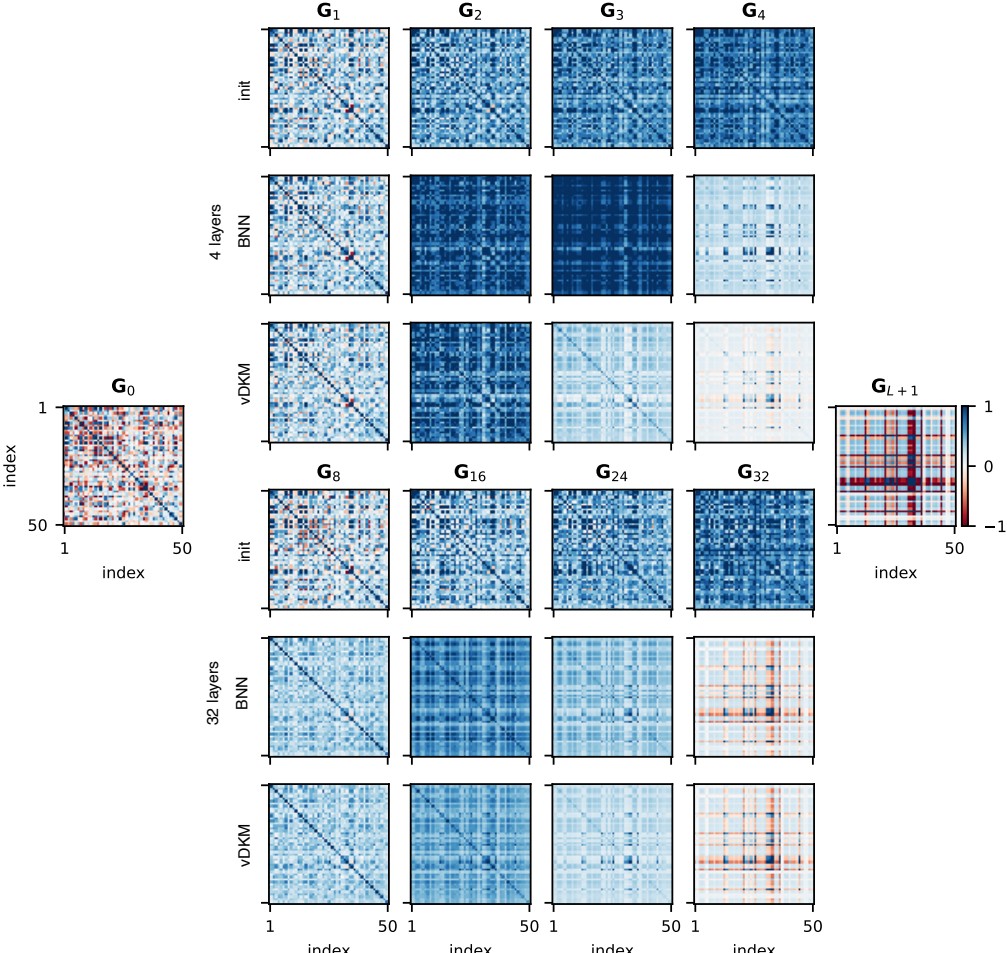

Figure 11: Comparison of Gram matrices between BNN of width 1024 (trained by Langevin sampling over features) and variational DKM, in 4-layer (row 1-3) and 32-layer networks (row 4-6). Initializations are shown in row 1 and 4, trained BNN Gram matrices are shown in row 2 and 5, and trained variational DKM Gram matrices are shown in row 3 and 6. As in Figure 10, the variational DKM is a poor match to Langevin sampling in a BNN for a 4-layer network, but is very similar in a 32 layer network.

# J    UNIMODALITY IN LINEAR DEEP KERNEL MACHINES

## J.1    THEORY: UNIMODALITY WITH A LINEAR KERNEL AND SAME WIDTHS

Here, we show that the deep kernel machine objective is unimodal for a linear kernel. A linear kernel simply returns the input Gram matrix,

$$\mathbf{K}(\mathbf{G}) = \mathbf{G}. \tag{114}$$

It is called a linear kernel, because it arises in the neural network setting (Eq. 21) by choosing the nonlinearity, $\phi$ to be the identity, in which case, $\mathbf{F}_\ell = \mathbf{F}_{\ell-1}\mathbf{W}_{\ell-1}$. For a linear kernel the objective becomes,

$$\mathcal{L}(\mathbf{G}_1, ..., \mathbf{G}_L) = \sum_{\ell=1}^{L+1} \tfrac{\nu_\ell}{2} \left( \log \left| \mathbf{G}_{\ell-1}^{-1}\mathbf{G}_\ell \right| - \mathrm{Tr}\left( \mathbf{G}_{\ell-1}^{-1}\mathbf{G}_\ell \right) \right) \tag{115}$$

where we have assumed there is no output noise, $\sigma^2 = 0$. Taking all $\nu_\ell$ to be equal, $\nu = \nu_\ell$ (see Appendix J.2 for the general case),

$$\mathcal{L}(\mathbf{G}_1, ..., \mathbf{G}_L) = \log \left| \mathbf{G}_0^{-1}\mathbf{G}_{L+1} \right| - \tfrac{\nu}{2} \sum_{\ell=1}^{L+1} \mathrm{Tr}\left( \mathbf{G}_{\ell-1}^{-1}\mathbf{G}_\ell \right). \tag{116}$$

Note that $\mathbf{G}_0$ and $\mathbf{G}_{L+1}$ are fixed by the inputs and outputs respectively. Thus, to find the mode, we set the gradient wrt $\mathbf{G}_1, \dots, \mathbf{G}_L$ to zero,

$$\mathbf{0} = \frac{\partial \mathcal{L}}{\partial \mathbf{G}_\ell} = \tfrac{\nu}{2} \left( \mathbf{G}_{\ell-1}^{-1} - \mathbf{G}_\ell^{-1}\mathbf{G}_{\ell+1}\mathbf{G}_\ell^{-1} \right) \tag{117}$$

Thus, at the mode, the recursive relationship must hold,

$$\mathbf{T} = \mathbf{G}_{\ell-1}^{-1}\mathbf{G}_\ell = \mathbf{G}_\ell^{-1}\mathbf{G}_{\ell+1}. \tag{118}$$

Thus, optimal Gram matrices are given by,

$$\mathbf{G}_\ell = \mathbf{G}_0\mathbf{T}^\ell, \tag{119}$$

and we can solve for $\mathbf{T}$ by noting,

$$\mathbf{G}_0^{-1}\mathbf{G}_{L+1} = \mathbf{T}^{L+1}. \tag{120}$$

Importantly, $\mathbf{T}$ is the product of two positive definite matrices, $\mathbf{T} = \mathbf{G}_{\ell-1}^{-1}\mathbf{G}_\ell$, so $\mathbf{T}$ must have positive, real eigenvalues (but $\mathbf{T}$ does not have to be symmetric (Horn & Johnson, 2012)). There is only one solution to Eq. (120) with positive real eigenvalues (Horn et al., 1994). Intuitively, this can be seen using the eigendecomposition, $\mathbf{G}_0^{-1}\mathbf{G}_{L+1} = \mathbf{V}^{-1}\mathbf{D}\mathbf{V}$, where $\mathbf{D}$ is diagonal,

$$\mathbf{T} = \left( \mathbf{V}^{-1}\mathbf{D}\mathbf{V} \right)^{1/(L+1)} = \mathbf{V}^{-1}\mathbf{D}^{1/(L+1)}\mathbf{V}. \tag{121}$$

Thus, finding $\mathbf{T}$ reduces to finding the $(L+1)$th root of each positive real number on the diagonal of $\mathbf{D}$. While there are $(L+1)$ complex roots, there is only one positive real root, and so $\mathbf{T}$ and hence $\mathbf{G}_1, \dots, \mathbf{G}_L$ are uniquely specified. This contrasts with a deep linear neural network, which has infinitely many optimal settings for the weights.

Note that for the objective to be well-defined, we need $\mathbf{K}(\mathbf{G})$ to be full-rank. With standard kernels (such as the squared exponential) this is always the case, even if the input Gram matrix is singular. However, a linear kernel will have a singular output if given a singular input, and with enough data points, $\mathbf{G}_0 = \frac{1}{\nu_0}\mathbf{X}\mathbf{X}^T$ is always singular. To fix this, we could e.g. define $\mathbf{G}_0 = \mathbf{K}(\frac{1}{\nu_0}\mathbf{X}\mathbf{X}^T)$ to be given by applying a positive definite kernel (such as a squared exponential) to $\frac{1}{\nu_0}\mathbf{X}\mathbf{X}^T$. This results in positive definite $\mathbf{G}_0$, as long as the input points are distinct.

## J.2    THEORY: UNIMODALITY WITH A LINEAR KERNEL AND ARBITRARY WIDTHS

In the main text we showed that the deep kernel machine is unimodal when all $\nu_\ell$ are equal. Here, we show that unimodality in linear DKMs also holds for all choices of $\nu_\ell$. Recall the linear DKM objective in Eq. (115),

$$\mathcal{L}(\mathbf{G}_1, ..., \mathbf{G}_L) = \sum_{\ell=1}^{L+1} \tfrac{\nu_\ell}{2} \left( \log \left| \mathbf{G}_{\ell-1}^{-1}\mathbf{G}_\ell \right| - \mathrm{Tr}\left( \mathbf{G}_{\ell-1}^{-1}\mathbf{G}_\ell \right) \right) \tag{122}$$

$$= \sum_{\ell=1}^{L+1} \tfrac{\nu_\ell}{2} \left( \log \left| \mathbf{G}_\ell \right| - \log \left| \mathbf{G}_{\ell-1} \right| - \mathrm{Tr}\left( \mathbf{G}_{\ell-1}^{-1}\mathbf{G}_\ell \right) \right). \tag{123}$$

To find the mode, again we set the gradient wrt $\mathbf{G}_\ell$ to zero,

$$0 = \frac{\partial \mathcal{L}}{\partial \mathbf{G}_\ell} = -\frac{\nu_{\ell+1}-\nu_\ell}{2}\mathbf{G}_\ell^{-1} - \frac{\nu_\ell}{2}\mathbf{G}_{\ell-1}^{-1} + \frac{\nu_{\ell+1}}{2}\mathbf{G}_\ell^{-1}\mathbf{G}_{\ell+1}\mathbf{G}_\ell^{-1}, \tag{124}$$

for $\ell = 1, ..., L$. Right multiplying by $2\mathbf{G}_\ell$ and rearranging,

$$\nu_{\ell+1}\mathbf{G}_\ell^{-1}\mathbf{G}_{\ell+1} = \nu_\ell\mathbf{G}_{\ell-1}^{-1}\mathbf{G}_\ell + (\nu_{\ell+1} - \nu_\ell)\,\mathbf{I}, \qquad \text{for } \ell = 1, ..., L. \tag{125}$$

Evaluating this expression for $\ell = 1$ and $\ell = 2$ gives,

$$\nu_2\mathbf{G}_1^{-1}\mathbf{G}_2 = \nu_1\mathbf{G}_0^{-1}\mathbf{G}_1 + (\nu_2 - \nu_1)\,\mathbf{I}, \tag{126}$$

$$\nu_3\mathbf{G}_2^{-1}\mathbf{G}_3 = \nu_2\mathbf{G}_1^{-1}\mathbf{G}_2 + (\nu_3 - \nu_2)\,\mathbf{I} = \nu_1\mathbf{G}_0^{-1}\mathbf{G}_1 + (\nu_3 - \nu_1)\,\mathbf{I}. \tag{127}$$

Recursing, we get,

$$\nu_\ell\mathbf{G}_{\ell-1}^{-1}\mathbf{G}_\ell = \nu_1\mathbf{G}_0^{-1}\mathbf{G}_1 + (\nu_\ell - \nu_1)\,\mathbf{I}. \tag{128}$$

Critically, this form highlights constraints on $\mathbf{G}_1$. In particular, the right hand side, $\mathbf{G}_{\ell-1}^{-1}\mathbf{G}_\ell$, is the product of two positive definite matrices, so has positive eigenvalues (but may be non-symmetric (Horn & Johnson, 2012)). Thus, all eigenvalues of $\nu_1\mathbf{G}_0^{-1}\mathbf{G}_1$ must be larger than $\nu_1 - \nu_\ell$, and this holds true at all layers. This will become important later, as it rules out inadmissible solutions.

Given $\mathbf{G}_0$ and $\mathbf{G}_1$, we can compute any $\mathbf{G}_\ell$ using,

$$\mathbf{G}_0^{-1}\mathbf{G}_\ell = \prod_{\ell'=1}^\ell \left(\mathbf{G}_{\ell'-1}^{-1}\mathbf{G}_{\ell'}\right) = \frac{1}{\prod_{\ell'=1}^\ell \nu_{\ell'}} \prod_{\ell'=1}^\ell \left(\nu_{\ell'}\mathbf{G}_{\ell'-1}^{-1}\mathbf{G}_{\ell'}\right) \tag{129}$$

$$\left(\prod_{\ell'=1}^\ell \nu_{\ell'}\right)\mathbf{G}_0^{-1}\mathbf{G}_\ell = \prod_{\ell'=1}^\ell \left(\nu_1\mathbf{G}_0^{-1}\mathbf{G}_1 + (\nu_{\ell'} - \nu_1)\,\mathbf{I}\right) \tag{130}$$

where the matrix products are ordered as $\prod_{\ell=1}^L \mathbf{A}_\ell = \mathbf{A}_1 \cdots \mathbf{A}_L$. Now, we seek to solve for $\mathbf{G}_1$ using our knowledge of $\mathbf{G}_{L+1}$. Computing $\mathbf{G}_0^{-1}\mathbf{G}_{L+1}$,

$$\left(\prod_{\ell=1}^{L+1} \nu_\ell\right)\mathbf{G}_0^{-1}\mathbf{G}_{L+1} = \prod_{\ell=1}^{L+1} \left(\nu_1\mathbf{G}_0^{-1}\mathbf{G}_1 + (\nu_\ell - \nu_1)\,\mathbf{I}\right). \tag{131}$$

We write the eigendecomposition of $\nu_1\mathbf{G}_0^{-1}\mathbf{G}_1$ as,

$$\nu_1\mathbf{G}_0^{-1}\mathbf{G}_1 = \mathbf{V}\mathbf{D}\mathbf{V}^{-1}. \tag{132}$$

Thus,

$$\left(\prod_{\ell=1}^{L+1} \nu_\ell\right)\mathbf{G}_0^{-1}\mathbf{G}_{L+1} = \prod_{\ell=1}^{L+1} \left(\mathbf{V}\mathbf{D}\mathbf{V}^{-1} + (\nu_\ell - \nu_1)\,\mathbf{I}\right) = \mathbf{V}\mathbf{\Lambda}\mathbf{V}^{-1} \tag{133}$$

where $\mathbf{\Lambda}$ is a diagonal matrix,

$$\mathbf{\Lambda} = \prod_{\ell=1}^{L+1} \left(\mathbf{D} + (\nu_\ell - \nu_1)\,\mathbf{I}\right). \tag{134}$$

Thus, we can identify $\mathbf{V}$ and $\mathbf{\Lambda}$ by performing an eigendecomposition of the known matrix, $\left(\prod_{\ell=1}^{L+1} \nu_\ell\right)\mathbf{G}_0^{-1}\mathbf{G}_{L+1}$. Then, we can solve for $\mathbf{D}$ (and hence $\mathbf{G}_1$) in terms of $\mathbf{\Lambda}$ and $\mathbf{V}$. The diagonal elements of $\mathbf{D}$ satisfy,

$$0 = -\Lambda_{ii} + \prod_{k=1}^{L+1} \left(D_{ii} + (\nu_\ell - \nu_1)\right). \tag{135}$$

This is a polynomial, and remembering the constraints from Eq. (128), we are interested in solutions which satisfy,

$$\nu_1 - \nu_{\min} \le D_{ii}. \tag{136}$$

where,

$$\nu_{\min} = \min\left(\nu_1, \ldots, \nu_{L+1}\right). \tag{137}$$

To reason about the number of such solutions, we use Descartes' rule of signs, which states that the number of positive real roots is equal to or a multiple of two less than the number of sign changes in the coefficients of the polynomial. Thus, if there is one sign change, there must be one positive real root. For instance, in the following polynomial,

$$0 = x^3 + x^2 - 1 \tag{138}$$

the signs go as $(+), (+), (-)$, so there is only one sign change, and there is one real root. To use Descartes' rule of signs, we work in terms of $D'_{ii}$, which is constrained to be positive,

$$0 \le D'_{ii} = D_{ii} - (\nu_1 - \nu_{\min}) \qquad\qquad D_{ii} = D'_{ii} + (\nu_1 - \nu_{\min}). \tag{139}$$

Thus, the polynomial of interest (Eq. 135) becomes,

$$0 = -\Lambda_{ii} + \prod_{\ell=1}^{L+1} \left(D'_{ii} + (\nu_1 - \nu_{\min}) - (\nu_1 - \nu_\ell)\right) = -\Lambda_{ii} + \prod_{\ell=1}^{L+1} \left(D'_{ii} + (\nu_\ell - \nu_{\min})\right) \tag{140}$$

where $0 < \nu_\ell - \nu_{\min}$ as $\nu_{\min}$ is defined to be the smallest $\nu_\ell$ (Eq. 137). Thus, the constant term, $-\Lambda_{ii}$ is negative, while all other terms, $D'_{ii}, \ldots, (D'_{ii})^{L+1}$ in the polynomial have positive coefficients. Thus, there is only one sign change, which proves the existence of only one valid real root, as required.

## K  UNIMODALITY EXPERIMENTS WITH NONLINEAR KERNELS

For the posterior over Gram matrices to converge to a point distribution, we need the DKM objective $\mathcal{L}(\mathbf{G}_1, \ldots, \mathbf{G}_L)$ to have one unique global optimum. As noted above, this is guaranteed when the prior dominates (Eq. 11), and for linear models (Appendix J). While we believe that it might be possible to construct counter examples, in practice we expect a single global optimum in most practical settings. To confirm this expectation, we did a number of experiments, starting with many different random initializations of a deep kernel machine and optimizing using gradient descent (Appendix K). In all cases tested, the optimizers converged to the same maximum.

We parameterise Gram matrices $\mathbf{G}_\ell = \frac{1}{P}\mathbf{V}_\ell\mathbf{V}_\ell^T$ with $\mathbf{V}_\ell \in \mathbb{R}^{P \times P}$ being trainable parameters. To make initializations with different seeds sufficiently separated while ensuring stability we initialize $\mathbf{G}_\ell$ from a broad distribution that depends on $\mathbf{K}(\mathbf{G}_{\ell-1})$. Specifically, we first take the Cholesky decomposition $\mathbf{K}(\mathbf{G}_{\ell-1}) = \mathbf{L}_{\ell-1}\mathbf{L}_{\ell-1}^T$, then set $\mathbf{V}_\ell = \mathbf{L}_{\ell-1}\mathbf{\Xi}_\ell\mathbf{D}_\ell^{1/2}$ where each entry of $\mathbf{\Xi}_\ell \in \mathbb{R}^{P \times P}$ is independently sampled from a standard Gaussian, and $\mathbf{D}_\ell$ is a diagonal scaling matrix with each entry sampled i.i.d. from an inverse-Gamma distribution. The variance of the inverse-Gamma distribution is fixed to 100, and the mean is drawn from a uniform distribution $U[0.5, 3]$ for each seed. Since for any random variable $x \sim \text{Inv-Gamma}(\alpha, \beta)$, $\mathbb{E}(x) = \frac{\beta}{\alpha-1}$ and $\mathbb{V}(x) = \frac{\beta}{(\alpha-1)(\alpha-2)}$, once we fix the mean and variance we can compute $\alpha$ and $\beta$ as

$$\alpha = \frac{\mathbb{E}(x)^2}{\mathbb{V}(x)} + 2, \tag{141}$$

$$\beta = \mathbb{E}(x)(\alpha - 1). \tag{142}$$

We set $\nu_\ell = 5$, and use the Adam optimizer (Kingma & Ba, 2014) with learning rate $0.001$ to optimize parameters $\mathbf{V}_\ell$ described above. We fixed all model hyperparameters to ensure that any multimodality could emerge only from the underlying deep kernel machine. As we did not use inducing points, we were forced to consider only the smaller UCI datasets (yacht, boston, energy and concrete). For the deep kernel machine objective, all Gram matrices converge rapidly to the same solution, as measured by RMSE (Fig. 12). Critically, we did find multiple modes for the MAP objective (Fig 13), indicating that experiments are indeed powerful enough to find multiple modes (though of course they cannot be guaranteed to find them). Finally, note that the Gram matrices took a surprisingly long time to converge: this was largely due to the high degree of diversity in the initializations; convergence was much faster if we initialised deterministically from the prior.

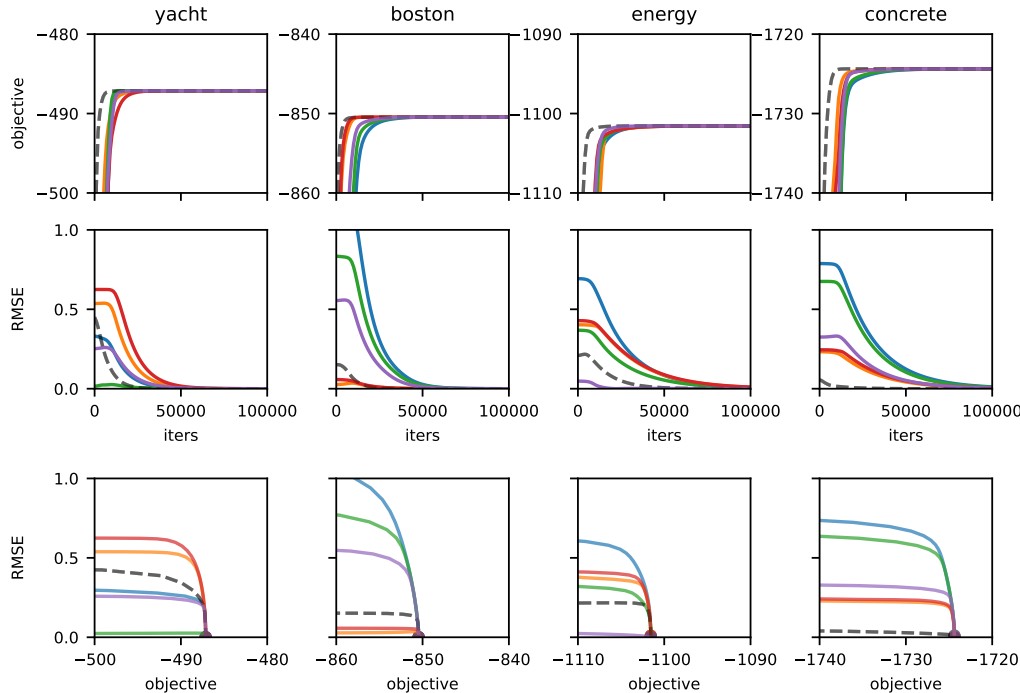

Figure 12: One-layer DKMs with squared exponential kernel trained on full UCI datasets (through columns) converges to the same solution, despite very different initializations by applying stochastic diagonal scalings described in Appendix F to the standard initialization with different seeds. Standard initialization is shown in dashed line, while scaled initializations are the color lines each denoting a different seed. The first row shows the objective during training for all seeds that all converge to the same value. The second row shows the element-wise RMSE between the Gram matrix of each seed and the optimized Gram matrix obtained from the standard initialization. RMSE converges to 0 as all initializations converge on the same maximum. The last row plots RMSE versus objective value, again showing a single optimal objective value where all Gram matrices are the same.

This might contradict our usual intuitions about huge multimodality in the weights/features of BNNs and DGPs. This can be reconciled by noting that each mode, written in terms of Gram matrices, corresponds to (perhaps infinitely) many modal features. In particular, in Sec. E, we show that the log-probability for features, $\mathrm{P}\left(\mathbf{F}_\ell | \mathbf{F}_{\ell-1}\right)$ (Eq. 82) depends only on the Gram matrices, and note that there are many settings of features which give the same Gram matrix. In particular, the Gram matrix is the same for any unitary transformation of the features, $\mathbf{F}'_\ell = \mathbf{F}_\ell \mathbf{U}$, satisfying $\mathbf{U}\mathbf{U}^T = \mathbf{I}$, as $\frac{1}{N_\ell}\mathbf{F}'_\ell \mathbf{F}'^T_\ell = \frac{1}{N_\ell}\mathbf{F}_\ell \mathbf{U}_\ell \mathbf{U}^T_\ell \mathbf{F}^T_\ell = \frac{1}{N_\ell}\mathbf{F}_\ell \mathbf{F}^T_\ell = \mathbf{G}_\ell$. For DGPs we can use any unitary matrix, so there are infinitely many sets of features consistent with a particular Gram matrix, while for BNNs we can only use permutation matrices, which are a subset of unitary matrices. Thus, the objective landscape must be far more complex in the feature domain than with Gram matrices, as a single optimal Gram matrix corresponds to a large family of optimal features.

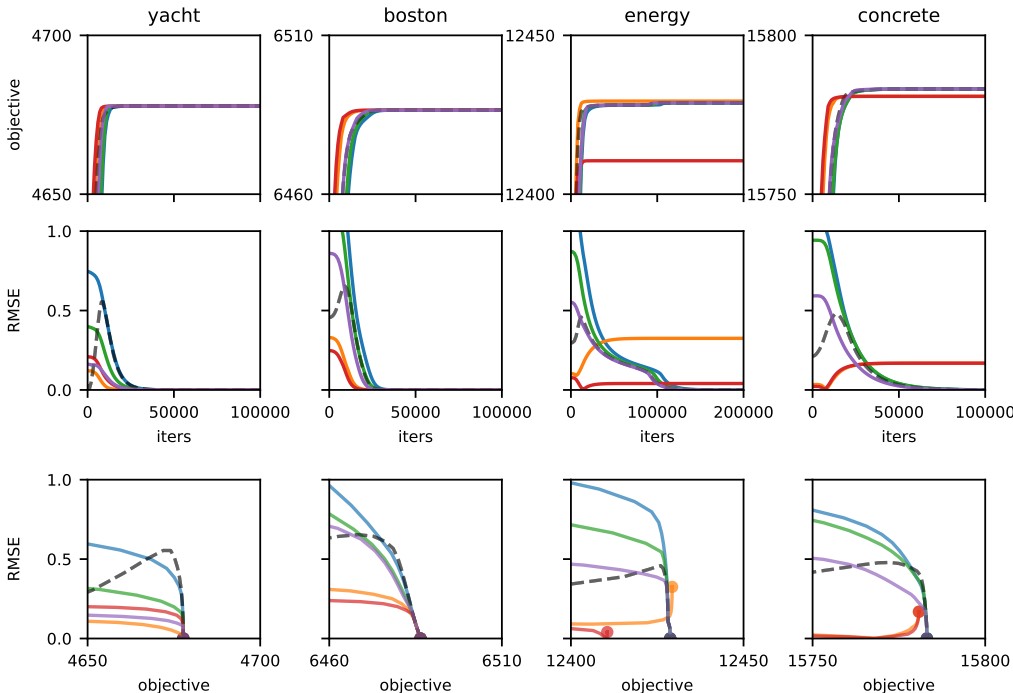

Figure 13: One-layer DGP with MAP inference over features as described in Appendix E Eq. (84). Rows and columns are the same as in Figure 12. Using the same randomly scaled initializations described above, we are able to find multiple modes in energy and concrete showing our initializations are diverse enough, albeit there is still only a single global optimum.

## L  INDUCING POINT DKMS

To do large-scale experiments on UCI datasets, we introduce inducing point DKMs by extending Gaussian process inducing point methods (Damianou & Lawrence, 2013; Salimbeni & Deisenroth, 2017) to the DKM setting. This approach uses the variational interpretation of the deep kernel machine objective described in Appendix D.

To do inducing-point variational inference, we need to explicitly introduce top-layer features mirroring $\mathbf{F}_{L+1} \in \mathbb{R}^{P \times \nu_{L+1}}$ in Appendix B, but replicated $N$ times, $\tilde{\mathbf{F}}_{L+1} \in \mathbb{R}^{P \times N_{L+1}}$. Formally, each feature, $\tilde{\mathbf{f}}_1^{L+1}, \ldots, \tilde{\mathbf{f}}_{N_{L+1}}^{L+1}$ is IID, conditioned on $\mathbf{F}_L$,

$$\mathrm{P}\left(\tilde{\mathbf{F}}_{L+1}|\mathbf{F}_L\right) = \prod_{\lambda=1}^{N_\ell} \mathcal{N}\left(\tilde{\mathbf{f}}_\lambda^{L+1}; \mathbf{0}, \mathbf{K}(\mathbf{G}(\mathbf{F}_L))\right), \tag{143a}$$

$$\mathrm{P}\left(\tilde{\mathbf{Y}}|\tilde{\mathbf{F}}_{L+1}\right) = \prod_{\lambda=1}^{N_\ell} \mathcal{N}\left(\tilde{\mathbf{y}}_\lambda; \tilde{\mathbf{f}}_\lambda^{L+1}, \sigma^2 \mathbf{I}\right), \tag{143b}$$

where we give the likelihood for regression, but other likelihoods (e.g. for classification) are possible (Appendix B).

Further, we take the total number of points, $P$, to be made up of $P_\mathrm{i}$ inducing points and $P_\mathrm{t}$ test/train points, so that $P = P_\mathrm{i} + P_\mathrm{t}$. Thus, we can separate all features, $\mathbf{F}_\ell \in \mathbb{R}^{P \times N_\ell}$, into the inducing features, $\mathbf{F}_\mathrm{i}^\ell \in \mathbb{R}^{P_\mathrm{i} \times N_\ell}$, and the test/train features, $\mathbf{F}_\mathrm{t}^\ell \in \mathbb{R}^{P_\mathrm{i} \times N_\ell}$. Likewise, we separate the inputs, $\mathbf{X}$, and outputs, $\mathbf{Y}$, into (potentially trained) inducing inputs, $\mathbf{X}_\mathrm{i}$, and trained inducing outputs, $\mathbf{Y}_\mathrm{i}$, and the real test/training inputs, $\mathbf{X}_\mathrm{t}$, and outputs, $\mathbf{Y}_\mathrm{t}$,

$$\mathbf{F}_\ell = \begin{pmatrix} \mathbf{F}_\mathrm{i}^\ell \\ \mathbf{F}_\mathrm{t}^\ell \end{pmatrix} \qquad \tilde{\mathbf{F}}_{L+1} = \begin{pmatrix} \tilde{\mathbf{F}}_\mathrm{i}^{L+1} \\ \tilde{\mathbf{F}}_\mathrm{t}^{L+1} \end{pmatrix} \qquad \mathbf{X} = \begin{pmatrix} \mathbf{X}_\mathrm{i} \\ \mathbf{X}_\mathrm{t} \end{pmatrix} \qquad \mathbf{Y} = \begin{pmatrix} \mathbf{Y}_\mathrm{i} \\ \mathbf{Y}_\mathrm{t} \end{pmatrix} \qquad \tilde{\mathbf{Y}} = \begin{pmatrix} \tilde{\mathbf{Y}}_\mathrm{i} \\ \tilde{\mathbf{Y}}_\mathrm{t} \end{pmatrix} \tag{144}$$

We follow the usual doubly stochastic inducing point approach for DGPs. In particular, we treat all the features at intermediate layers, $\mathbf{F}_1, \ldots, \mathbf{F}_L$, and the top-layer train/test features, $\mathbf{F}_\mathrm{t}^{L+1}$ as latent variables. However, we deviate from the usual setup in treating the top-layer inducing outputs, $\mathbf{F}_\mathrm{i}^{L+1}$, as learned parameters and maximize over them to ensure that the ultimate method does not require sampling, and at the same time allows minibatched training. The prior and approximate posterior over $\mathbf{F}_1, \ldots, \mathbf{F}_L$ are given by,

$$\mathrm{Q}\left(\mathbf{F}_1, \ldots \mathbf{F}_L | \mathbf{X}\right) = \prod_{\ell=1}^L \mathrm{Q}\left(\mathbf{F}_\ell | \mathbf{F}_{\ell-1}\right), \tag{145a}$$

$$\mathrm{P}\left(\mathbf{F}_1, \ldots, \mathbf{F}_L | \mathbf{X}\right) = \prod_{\ell=1}^L \mathrm{P}\left(\mathbf{F}_\ell | \mathbf{F}_{\ell-1}\right), \tag{145b}$$

and remember $\mathbf{F}_0 = \mathbf{X}$, so $\mathbf{G}_0 = \frac{1}{N_0}\mathbf{X}\mathbf{X}^T$. The prior and approximate posterior at each layer factorises into a distribution over the inducing points and a distribution over the test/train points,

$$\mathrm{Q}\left(\mathbf{F}_\ell | \mathbf{F}_{\ell-1}\right) = \mathrm{P}\left(\mathbf{F}_\mathrm{t}^\ell | \mathbf{F}_\mathrm{i}^\ell, \mathbf{F}_{\ell-1}\right) \mathrm{Q}\left(\mathbf{F}_\mathrm{i}^\ell\right), \tag{146a}$$

$$\mathrm{P}\left(\mathbf{F}_\ell | \mathbf{F}_{\ell-1}\right) = \mathrm{P}\left(\mathbf{F}_\mathrm{t}^\ell | \mathbf{F}_\mathrm{i}^\ell, \mathbf{F}_{\ell-1}\right) \mathrm{P}\left(\mathbf{F}_\mathrm{i}^\ell | \mathbf{F}_\mathrm{i}^{\ell-1}\right). \tag{146b}$$

Critically, the approximate posterior samples for the test/train points is the conditional prior $\mathrm{P}\left(\mathbf{F}_\mathrm{t}^\ell | \mathbf{F}_\mathrm{i}^\ell, \mathbf{F}_{\ell-1}\right)$, which is going to lead to cancellation when we compute the ELBO. Likewise, the approximate posterior over $\tilde{\mathbf{F}}_\mathrm{t}^{L+1}$ is the conditional prior,

$$\mathrm{Q}\left(\tilde{\mathbf{F}}_\mathrm{t}^{L+1} | \mathbf{F}_\mathrm{i}^{L+1}, \mathbf{F}_L\right) = \mathrm{P}\left(\tilde{\mathbf{F}}_\mathrm{t}^{L+1} | \mathbf{F}_\mathrm{i}^{L+1}, \mathbf{F}_L\right). \tag{147}$$

Concretely, the prior approximate posterior over inducing points are given by,

$$\mathrm{Q}\left(\mathbf{F}_\mathrm{i}^\ell\right) = \prod_{\lambda=1}^{N_\ell} \mathcal{N}\left(\mathbf{f}_{\mathrm{i};\lambda}^\ell; \mathbf{0}, \mathbf{G}_{\mathrm{ii}}^\ell\right), \tag{148a}$$

$$\mathrm{P}\left(\mathbf{F}_\mathrm{i}^\ell | \mathbf{F}_\mathrm{i}^{\ell-1}\right) = \prod_{\lambda=1}^{N_\ell} \mathcal{N}\left(\mathbf{f}_{\mathrm{i};\lambda}^\ell; \mathbf{0}, \mathbf{K}(\mathbf{G}(\mathbf{F}_\mathrm{i}^{\ell-1}))\right) \tag{148b}$$

The approximate posterior is directly analogous to Eq. (69) and the prior is directly analogous to Eq. (1a), but where we have specified that this is only over inducing points. Now we compute the ELBO

$$\mathrm{ELBO}(\mathbf{F}_\mathrm{i}^{L+1}, \mathbf{G}_{\mathrm{ii}}^1, \ldots, \mathbf{G}_{\mathrm{ii}}^L)$$

$$= \mathbb{E}_\mathrm{Q}\left[\log \mathrm{P}\left(\tilde{\mathbf{Y}}_\mathrm{t} | \tilde{\mathbf{F}}_\mathrm{t}^{L+1}\right) + \log \frac{\mathrm{P}\left(\tilde{\mathbf{F}}_\mathrm{t}^{L+1} | \mathbf{F}_\mathrm{i}^{L+1}, \mathbf{F}_L\right) \mathrm{P}\left(\mathbf{F}_1, \ldots \mathbf{F}_L | \mathbf{X}\right)}{\mathrm{Q}\left(\tilde{\mathbf{F}}_\mathrm{t}^{L+1} | \mathbf{F}_\mathrm{i}^{L+1}, \mathbf{F}_L\right) \mathrm{Q}\left(\mathbf{F}_1, \ldots \mathbf{F}_L | \mathbf{X}\right)}\right] \tag{149}$$

Note that the P $\left(\mathbf{F}_\mathrm{t}^\ell | \mathbf{F}_\mathrm{i}^\ell, \mathbf{F}_{\ell-1}\right)$ terms are going to cancel in the ELBO, we consider them below when we come to describing sampling), substituting Eq. (145–147) and cancelling P $\left(\mathbf{F}_\mathrm{t}^\ell | \mathbf{F}_\mathrm{i}^\ell, \mathbf{F}_{\ell-1}\right)$ and P $\left(\tilde{\mathbf{F}}_\mathrm{t}^{L+1} | \mathbf{F}_\mathrm{i}^{L+1}, \mathbf{F}_L\right)$,

$$\text{ELBO}(\mathbf{F}_\mathrm{i}^{L+1}, \mathbf{G}_\mathrm{ii}^1, \ldots, \mathbf{G}_\mathrm{ii}^L) = \mathbb{E}_Q \left[ \log \mathrm{P}\left(\tilde{\mathbf{Y}}_\mathrm{t} | \tilde{\mathbf{F}}_\mathrm{t}^{L+1}\right) + \sum_{\ell=1}^L \log \frac{\mathrm{P}\left(\mathbf{F}_\mathrm{i}^\ell | \mathbf{F}_\mathrm{i}^{\ell-1}\right)}{\mathrm{Q}\left(\mathbf{F}_\mathrm{i}^\ell\right)} \right]. \tag{150}$$

So far, we have treated the Gram matrices, $\mathbf{G}_\mathrm{ii}^\ell$ as parameters of the approximate posterior. However, in the infinite limit $N \to \infty$, these are consistent with the features generated by the approximate posterior. In particular the matrix product $\frac{1}{N_\ell} \mathbf{F}_\mathrm{i}^\ell \left(\mathbf{F}_\mathrm{i}^\ell\right)^T$ can be written as an average over infinitely many IID vectors, $\mathbf{f}_{\mathrm{i};\lambda}^\ell$ (first equality), and by the law of large numbers, this is equal to the expectation of one term (second equality), which is $\mathbf{G}_\mathrm{ii}^\ell$ (by the approximate posterior Eq. (148a)),

$$\lim_{N \to \infty} \tfrac{1}{N_\ell} \mathbf{F}_\mathrm{i}^\ell \left(\mathbf{F}_\mathrm{i}^\ell\right)^T = \lim_{N \to \infty} \tfrac{1}{N_\ell} \sum_{\lambda=1}^{N_\ell} \mathbf{f}_{\mathrm{i};\lambda}^\ell \left(\mathbf{f}_{\mathrm{i};\lambda}^\ell\right)^T = \mathbb{E}_{\mathrm{Q}\left(\mathbf{f}_{\mathrm{i};\lambda}^\ell\right)} \left[ \mathbf{f}_{\mathrm{i};\lambda}^\ell \left(\mathbf{f}_{\mathrm{i};\lambda}^\ell\right)^T \right] = \mathbf{G}_\mathrm{ii}^\ell. \tag{151}$$

By this argument, the Gram matrix from the previous layer, $\mathbf{G}_\mathrm{ii}^{\ell-1}$ is deterministic. Further, in a DGP, $\mathbf{F}_\mathrm{i}^\ell$ only depends on $\mathbf{F}_\mathrm{i}^{\ell-1}$ through $\mathbf{G}_\mathrm{i}^{\ell-1}$ (Eq. 5), and the prior and approximate posterior factorise. Thus, in the infinite limit, individual terms in the ELBO can be written,

$$\lim_{N \to \infty} \tfrac{1}{N} \mathbb{E}_Q \left[ \log \frac{\mathrm{P}\left(\mathbf{F}_\mathrm{i}^\ell | \mathbf{F}_\mathrm{i}^{\ell-1}\right)}{\mathrm{Q}\left(\mathbf{F}_\mathrm{i}^\ell\right)} \right] = \nu_\ell \, \mathbb{E}_Q \left[ \log \frac{\mathrm{P}\left(\mathbf{f}_{\mathrm{i};\lambda}^\ell | \mathbf{G}_\mathrm{i}^{\ell-1}\right)}{\mathrm{Q}\left(\mathbf{f}_{\mathrm{i};\lambda}^\ell\right)} \right] \tag{152}$$

$$= -\nu_\ell \, \mathrm{D}_\mathrm{KL} \left( \mathcal{N}\left(\mathbf{0}, \mathbf{K}(\mathbf{G}_\mathrm{i}^{\ell-1})\right) \big\| \mathcal{N}\left(\mathbf{0}, \mathbf{G}_\mathrm{i}^\ell\right)\right), \tag{153}$$

where the final equality arises when we notice that the expectation can be written as a KL-divergence. The inducing DKM objective, $\mathcal{L}_\mathrm{ind}$, is the ELBO, divided by N to ensure that it remains finite in the infinite limit,

$$\mathcal{L}_\mathrm{ind}(\mathbf{F}_\mathrm{i}^{L+1}, \mathbf{G}_\mathrm{ii}^1, \ldots, \mathbf{G}_\mathrm{ii}^L) = \lim_{N \to \infty} \tfrac{1}{N} \text{ELBO}(\mathbf{F}_\mathrm{i}^{L+1}, \mathbf{G}_\mathrm{ii}^1, \ldots, \mathbf{G}_\mathrm{ii}^L) \tag{154}$$

$$= \mathbb{E}_Q \left[ \log \mathrm{P}\left(\mathbf{Y}_\mathrm{t} | \mathbf{F}_\mathrm{t}^{L+1}\right) \right] - \sum_{\ell=1}^L \nu_\ell \, \mathrm{D}_\mathrm{KL} \left( \mathcal{N}\left(\mathbf{0}, \mathbf{K}(\mathbf{G}_\mathrm{ii}^{\ell-1})\right) \big\| \mathcal{N}\left(\mathbf{0}, \mathbf{G}_\mathrm{ii}^\ell\right)\right).$$

Note that this has almost exactly the same form as the standard DKM objective for DGPs in the main text (Eq. 16). In particular, the second term is a chain of KL-divergences, with the only difference that these KL-divergences apply only to the inducing points. The first term is a "performance" term that here depends on the quality of the predictions given the inducing points. As the copies are IID, we have,

$$\mathbb{E}_Q \left[ \log \mathrm{P}\left(\tilde{\mathbf{Y}}_\mathrm{t} | \tilde{\mathbf{F}}_\mathrm{t}^{L+1}\right) \right] = N \, \mathbb{E}_Q \left[ \log \mathrm{P}\left(\mathbf{Y}_\mathrm{t} | \mathbf{F}_\mathrm{t}^{L+1}\right) \right]. \tag{155}$$

Now that we have a simple form for the ELBO, we need to compute the expected likelihood, $\mathbb{E}_Q \left[ \log \mathrm{P}\left(\mathbf{Y}_\mathrm{t} | \mathbf{F}_\mathrm{t}^{L+1}\right) \right]$. This requires us to compute the full Gram matrices, including test/train points, conditioned on the optimized inducing Gram matrices. We start by defining the full Gram matrix,

$$\mathbf{G}_\ell = \begin{pmatrix} \mathbf{G}_\mathrm{ii}^\ell & \mathbf{G}_\mathrm{it}^\ell \\ \mathbf{G}_\mathrm{ti}^\ell & \mathbf{G}_\mathrm{tt}^\ell \end{pmatrix} \tag{156}$$

for both inducing points (labelled "i") and test/training points (labelled "t") from just $\mathbf{G}_\mathrm{ii}^\ell$. For clarity, we have $\mathbf{G}_\ell \in \mathbb{R}^{P \times P}$, $\mathbf{G}_\mathrm{ii}^\ell \in \mathbb{R}^{P_\mathrm{i} \times P_\mathrm{i}}$, $\mathbf{G}_\mathrm{ti}^\ell \in \mathbb{R}^{P_\mathrm{t} \times P_\mathrm{i}}$, $\mathbf{G}_\mathrm{tt}^\ell \in \mathbb{R}^{P_\mathrm{t} \times P_\mathrm{t}}$, where $P_\mathrm{i}$ is the number of inducing points, $P_\mathrm{t}$ is the number of train/test points and $P = P_\mathrm{i} + P_\mathrm{t}$ is the total number of inducing and train/test points.

The conditional distribution over $\mathbf{F}_\mathrm{t}^\ell$ given $\mathbf{F}_\mathrm{i}^\ell$ is,

$$\mathrm{P}\left(\mathbf{F}_\mathrm{t}^\ell | \mathbf{F}_\mathrm{i}^\ell, \mathbf{G}_{\ell-1}\right) = \prod_{\lambda=1}^{N_\ell} \mathcal{N}\left(\mathbf{f}_{\mathrm{t};\lambda}^\ell; \mathbf{K}_\mathrm{ti} \mathbf{K}_\mathrm{ii}^{-1} \mathbf{f}_{\mathrm{i};\lambda}^\ell, \mathbf{K}_{\mathrm{tt}\cdot\mathrm{i}}\right) \tag{157}$$

---

**Algorithm 1** DKM prediction

---

**Parameters:** $\{\nu_\ell\}_{\ell=1}^L$
**Optimized Gram matrices** $\{\mathbf{G}_{ii}^\ell\}_{\ell=1}^L$
**Inducing and train/test inputs:** $\mathbf{X}_i$, $\mathbf{X}_t$
**Inducing outputs:** $\mathbf{F}_i^{L+1}$
Initialize full Gram matrix
$$\begin{pmatrix} \mathbf{G}_{ii}^0 & \mathbf{G}_{ti}^{0;T} \\ \mathbf{G}_{ti}^0 & \mathbf{G}_{tt}^0 \end{pmatrix} = \tfrac{1}{\nu_0} \begin{pmatrix} \mathbf{X}_i\mathbf{X}_i^T & \mathbf{X}_i\mathbf{X}_t^T \\ \mathbf{X}_t\mathbf{X}_i^T & \mathbf{X}_t\mathbf{X}_t^T \end{pmatrix}$$
Propagate full Gram matrix
**for** $\ell$ **in** $(1,\dots,L)$ **do**
$$\begin{pmatrix} \mathbf{K}_{ii} & \mathbf{K}_{ti}^T \\ \mathbf{K}_{ti} & \mathbf{K}_{tt} \end{pmatrix} = \mathbf{K}\left(\begin{pmatrix} \mathbf{G}_{ii}^{\ell-1} & (\mathbf{G}_{ti}^{\ell-1})^T \\ \mathbf{G}_{ti}^{\ell-1} & \mathbf{G}_{tt}^{\ell-1} \end{pmatrix}\right)$$
$\quad \mathbf{K}_{tt\cdot i} = \mathbf{K}_{tt} - \mathbf{K}_{ti}\mathbf{K}_{ii}^{-1}\mathbf{K}_{ti}^T.$
$\quad \mathbf{G}_{ti}^\ell = \mathbf{K}_{ti}\mathbf{K}_{ii}^{-1}\mathbf{G}_{ii}^\ell$
$\quad \mathbf{G}_{tt}^\ell = \mathbf{K}_{ti}\mathbf{K}_{ii}^{-1}\mathbf{G}_{ii}^\ell\mathbf{K}_{ii}^{-1}\mathbf{K}_{ti}^T + \mathbf{K}_{tt\cdot i}$
**end for**
Final prediction using standard Gaussian process expressions
$$\begin{pmatrix} \mathbf{K}_{ii} & \mathbf{K}_{ti}^T \\ \mathbf{K}_{ti} & \mathbf{K}_{tt} \end{pmatrix} = \mathbf{K}\left(\begin{pmatrix} \mathbf{G}_{ii}^L & (\mathbf{G}_{ti}^L)^T \\ \mathbf{G}_{ti}^L & \mathbf{G}_{tt}^L \end{pmatrix}\right)$$
$\mathbf{Y}_t \sim \mathcal{N}\left(\mathbf{K}_{ti}\mathbf{K}_{ii}^{-1}\mathbf{F}_i^{L+1}, \mathbf{K}_{tt} - \mathbf{K}_{ti}\mathbf{K}_{ii}^{-1}\mathbf{K}_{ti}^T + \sigma^2\mathbf{I}\right)$

---

where $\mathbf{f}_{t;\lambda}^\ell$ is the activation of the $\lambda$th feature for all train/test inputs, $\mathbf{f}_{i;\lambda}^\ell$ is the activation of the $\lambda$th feature for all train/test inputs, and $\mathbf{f}_{i;\lambda}^\ell$, and

$$\begin{pmatrix} \mathbf{K}_{ii} & \mathbf{K}_{ti}^T \\ \mathbf{K}_{ti} & \mathbf{K}_{tt} \end{pmatrix} = \mathbf{K}\left(\tfrac{1}{N_{\ell-1}}\mathbf{F}_{\ell-1}\mathbf{F}_{\ell-1}^T\right) = \mathbf{K}\left(\mathbf{G}_{\ell-1}\right) \tag{158}$$

$$\mathbf{K}_{tt\cdot i} = \mathbf{K}_{tt} - \mathbf{K}_{ti}\mathbf{K}_{ii}^{-1}\mathbf{K}_{ti}^T. \tag{159}$$

In the infinite limit, the Gram matrix becomes deterministic via the law of large numbers (as in Eq. 151), and as such $\mathbf{G}_{it}$ and $\mathbf{G}_{tt}$ become deterministic and equal to their expected values. Using Eq. (157), we can write,

$$\mathbf{F}_t^\ell = \mathbf{K}_{ti}\mathbf{K}_{ii}^{-1}\mathbf{F}_i^\ell + \mathbf{K}_{tt\cdot i}^{1/2}\mathbf{\Xi}. \tag{160}$$

where $\mathbf{\Xi}$ is a matrix with IID standard Gaussian elements. Thus,

$$\mathbf{G}_{ti}^\ell = \tfrac{1}{\nu}\,\mathbb{E}\left[\mathbf{F}_t^\ell(\mathbf{F}_i^\ell)^T\right] \tag{161}$$

$$= \tfrac{1}{\nu}\mathbf{K}_{ti}\mathbf{K}_{ii}^{-1}\,\mathbb{E}\left[\mathbf{F}_i^\ell(\mathbf{F}_i^\ell)^T\right] \tag{162}$$

$$= \mathbf{K}_{ti}\mathbf{K}_{ii}^{-1}\mathbf{G}_{ii}^\ell \tag{163}$$

and,

$$\mathbf{G}_{tt}^\ell = \tfrac{1}{\nu}\,\mathbb{E}\left[\mathbf{F}_t^\ell(\mathbf{F}_t^\ell)^T\right] \tag{164}$$

$$= \tfrac{1}{\nu}\mathbf{K}_{ti}\mathbf{K}_{ii}^{-1}\,\mathbb{E}\left[\mathbf{F}_i^\ell(\mathbf{F}_i^\ell)^T\right]\mathbf{K}_{ii}^{-1}\mathbf{K}_{ti}^T + \tfrac{1}{\nu}\mathbf{K}_{tt\cdot i}^{1/2}\,\mathbb{E}\left[\mathbf{\Xi}\mathbf{\Xi}^T\right]\mathbf{K}_{tt\cdot i}^{1/2} \tag{165}$$

$$= \mathbf{K}_{ti}\mathbf{K}_{ii}^{-1}\mathbf{G}_{ii}\mathbf{K}_{ii}^{-1}\mathbf{K}_{ti}^T + \mathbf{K}_{tt\cdot i} \tag{166}$$

For the full prediction algorithm, see Alg. 1.

