# OpenReview forum: "A theory of representation learning in neural networks gives a deep generalisation of kernel methods"
_ICLR.cc/2023/Conference — Submitted to ICLR 2023_

### Official Review · Reviewer_SEBy · 2022-10-19

**Confidence:** 5
**Correctness:** 4
**Technical Novelty And Significance:** 2
**Empirical Novelty And Significance:** 2
**Recommendation:** 3

**Clarity, Quality, Novelty And Reproducibility:**

The paper is well-written. However, as explained above, the main contribution of this article is not original.

**Strength And Weaknesses:**

Strengths:
- The paper is well-written
- Understanding feature learning in Bayesian inference in mean-field networks is certainly an interesting topic
- The article points out a nice closed form solution for the MAP value of the feature-feature covariances for deep linear networks in the mean-field regime. I believe this point is novel.

Weaknesses:
- The main contribution of this article has been considered in many prior articles on mean-field neural networks, though it is phrased somewhat differently (in terms of rescaling the likelihood instead of the weight variance in the final layer). Below are a few representative references. The article would be significantly strengthened by connecting to this literature.

Yang, Greg, and Edward J. Hu. "Tensor programs iv: Feature learning in infinite-width neural networks." International Conference on Machine Learning. PMLR, 2021.

Sirignano, Justin, and Konstantinos Spiliopoulos. "Mean field analysis of neural networks: A central limit theorem." Stochastic Processes and their Applications 130.3 (2020): 1820-1852.

Sirignano, Justin, and Konstantinos Spiliopoulos. "Mean field analysis of neural networks: A law of large numbers." SIAM Journal on Applied Mathematics 80.2 (2020): 725-752.

Mei, Song, Andrea Montanari, and Phan-Minh Nguyen. "A mean field view of the landscape of two-layer neural networks." Proceedings of the National Academy of Sciences 115.33 (2018): E7665-E7671.

Nguyen, Phan-Minh, and Huy Tuan Pham. "A rigorous framework for the mean field limit of multilayer neural networks." arXiv preprint arXiv:2001.11443 (2020).

**Summary Of The Paper:**

This article considers Bayesian inference with fully connected networks of depth L and width n in the infinite width limit. In order to have non-trivial feature learning at infinite width, the authors propose scaling the MSE negative log-likelihood by a factor proportional to the width, causing it to be appear at the same order in n relative to the prior (written  in terms of the feature-feature covariances $G^\ell$ of layers ($\ell = 1,…,L$). The authors explicitly find the MAP solution for deep linear networks and provide some numerical investigation.

While the paper is clearly written, the authors don’t seem to realize that the likelihood re-scaling they propose is precisely what is called the mean-field scaling of neural networks in which the variance of final layer weights is rescaled by an extra factor or 1 / n (see below for references). In the language of this article, this precisely corresponds to rescaling the prior $P( Y | G^L )$ by a factor of n since it multiplies the variance of Y given $G^L$ by an additional factor of 1 / n. Given the significant overlap with existing work, I believe it is not ready to be published.

**Summary Of The Review:**

This article proposes in a Bayesian framework a well-known modification of neural networks, the so-called mean field regime, that makes them capable of learning features at infinite width and any fixed depth. The article also provides an explicit form of the MAP feature-feature covariances for deep linear networks. While a more detailed understanding of Bayesian interference in the mean-field regime is certainly of interest, the article is not sufficiently novel to be published in its current form.

---

> ### Author Response · Authors · 2022-11-10
> **Response**
>
> The reviewer suggests that our work lacks any novelty at all, based on a proposed equivalence.
> In particular, the reviewer proposes that sending the covariance to zero for a few outputs (as in past work) should be equivalent to scaling the likelihood by replicating the data (our approach).
> Formally, this would seem to translate to the claim that $N \log \mathcal{N}(Y; 0, K(G_L))$ equals $\log \mathcal{N}(Y; 0, K(G_L)/N)$, where $\mathcal{N}(Y; 0, K)$ is probability density of $Y$ under a multivariate Gaussian distribution with mean $0$ and covariance $K$.
> However, we can easily show that these quantities are not equal,
> \begin{align}
> N \log \mathcal{N}(Y; 0, K(G_L)) \neq \log \mathcal{N}(Y; 0, K(G_L)/N).
> \end{align}
> In particular, our scaled Gaussian likelihood is,
> \begin{align}
> N \log \mathcal{N}(Y; 0, K(G_L)) = -N \tfrac{\nu_L}{2} \log |K(G_L)| - \tfrac{N}{2} Y K^{-1}(G_L) Y^T.
> \end{align}
> In contrast, scaling the covariance gives,
> \begin{align}
> \log N(Y; 0, K(G_L)/N) &= -\tfrac{\nu_L}{2} \log |K(G_L)/N| - \tfrac{N}{2} Y K^{-1}(G_L) Y^T,
> \end{align}
> \begin{align}
> \log N(Y; 0, K(G_L)/N) &= -\tfrac{\nu_L}{2} \log |K(G_L)| +\tfrac{\nu_L P}{2} \log N - \tfrac{N}{2} Y K^{-1}(G_L) Y^T.
> \end{align}
> Thus, while the quadratic terms are equal, the log-determinant normalizer terms are not.
> And of course, the normalizer cannot be neglected as it plays an important part in shaping the top-layer representation, $G_L$.
>
> The lack of this particular equivalence is really a moot point, given the numerous dramatic differences to the cited past work.  We have written a contributions section, and rewritten the related work so as to highlight these undisputed contributions.  In particular, we show that:
> * We present a new infinite limit, the Bayesian representation learning limit, suitable for deep Bayesian models including DGPs.  This limit is novel (see above argument, and reviewer gVJs).
> * We show that in the Bayesian representation learning limit, DGP posteriors are exactly zero-mean multivariate Gaussian.
> * We show that the posterior covariances can be obtained by optimizing the "deep kernel machine objective", which combines a log-likelihood with a series of KL-divergences of multivariate Gaussians.
> * We give an interpretation of this objective, with $\log P(Y| G_L)$ encouraging improved performance, while the KL-divergence terms act as a regulariser, keeping representations close to those under the deterministic, infinite prior.
> * We introduce a sparse DKM, which takes inspiration GP inducing point literature to obtain a practical, scalable method that is linear in the number of datapoints. In contrast, naively computing/optimizing the DKM objective is cubic in the number of datapoints (as with most other naive kernel methods).

---

### Official Review · Reviewer_gVJs · 2022-10-24

**Confidence:** 3
**Correctness:** 3
**Technical Novelty And Significance:** 4
**Empirical Novelty And Significance:** Not applicable
**Recommendation:** 6

**Clarity, Quality, Novelty And Reproducibility:**

The proposed modification to the Bayesian algorithm is novel/original. However, the paper was not written in a clear manner. Here are a few suggestions (though far from exhaustive).

I. On the core trick: the core trick of (noticing diminishing effects of target labels on posterior hidden-layer representations and then amplifying them by) replicating labels can be stated simply, yet it is not clear until a reader gets to the paragraph surrounding Eq.(16) buried in Sec.3.3. It would be beneficial/motivating to have it explained up front in Section 1.

II. On large datasets: since the paper makes a claim about scalability of the algorithm to large datasets, it would be beneficial to (i) have a discussion of how the computational time scales with the training dataset size and (ii) to detail how many training samples there are in each training dataset used in the paper.

**Strength And Weaknesses:**

As stated in the "Summary Of The Paper" section, this paper contributes a nontrivial idea (strength) -- the construction of the Bayesian representation learning limit through label replications -- albeit with the overstated claim on the theoretical tractability (weakness).

Specifically, the paper claims that the proposed Bayesian algorithm is "extremely theoretically tractable." While it is easy to state the optimization objective, its theoretical analysis doesn't seem to be tractable for general deep neural networks (which is not surprising as the maximal-update parametrization is also hard to theoretically analyze after some gradient-descent iterations). In fact, as explained in the paper itself, to solve for the maximum of the objective in Eq.(17)/Eq.(21), in general one must resort to numerical/variational methods.

Similarly (but in an opposite direction), the large body of work listed in the second paragraph of Section 2 -- which utilizes perturbation theory to analyze large-but-finite-width neural networks -- is dismissed as "highly complex." However, in these perturbative approaches, one can systematically understand the behavior of networks order by order in 1/width, and get layer-by-layer recursive equations -- much like that for the NNGP in the infinite-width limit -- that determine the depth dependences of these perturbative corrections. This is theoretically analyzable -- hence such a large body of work exists -- and dismissing them as "highly complex" seems misleading (especially because the paper uses such a claim to further illustrate the simplicity of the proposed representation learning limit).

All those said, the proposed Bayesian algorithm remains nontrivial and interesting. But these claims on the theoretical (in)tractability should be clarified/softened/dropped, unless there is a substantive counter argument.


Minor comments:

A. It is preferable _not_ to use subjective adjectives/adverbs. For instance:

(A-i) "elegant objective" in the abstract: it is not clear what makes Eqs.(17)/(21) "elegant."

(A-ii) "strong regulariser" in 3.4: it is not clear what makes it "strong" (in fact, the relative width ratio, \nu_{\ell}=N_{\ell}/N, in front of each KL-divergence term controls its strength, so the relative strength of the regularizer can be adjusted).

(A-iii) "extremely theoretically tractable" in the abstract: "extremely" is subjective (even if we temporarily forget the previous major objection to this claim).

B. A missing period for "which we call the deep kernel machine (DKM) Here" in 3.7. (Also, the terminology/abbreviation is already introduced long before and used throughout the paper, so this phrasing doesn't make sense here.)

C. In the first paragraph, for "(NNGP Matthews et al., 2018; Novak et al., 2018)" it is probably more standard to cite [Lee et al., 2017] as well (or in place of [Novak et al., 2018]). (I am not an author of these papers.)

D. It may be appropriate to also cite [Dyer and Gur-Ari, arXiv:1909.11304] and [Hanin and Nica, arXiv:1909.05989] in the second paragraph of Section 2. (I am not an author of these papers.)



**Summary Of The Paper:**

This paper introduces a Bayesian learning algorithm that preserves the representation-learning ability of neural networks in the fixed-depth-infinite-width limit. This result parallels that of the maximal-update parametrization [Yang & Hu (2020)], which attains the similar limit in the gradient-descent setup (in contrast to the Bayesian setup discussed herein). In order to construct such a limit, the paper proposes to amplify the effect of target labels on the posterior representation by replicating labels N times, where N is a typical width of hidden layers. The proposed algorithm is applied to several tasks.

**Summary Of The Review:**

Overall, the recommendation for "marginal acceptance" is based on the novelty and possible significance of the proposed algorithm on deep learning theory. The weakness should be addressed.

---

> ### Author Response · Authors · 2022-11-10
> **Response**
>
> Thanks for your considered and positive review!  Please note that we have made extensive changes in response to your comments, but that changes to the abstract appear only in the pdf.
>
> > Specifically, the paper claims that the proposed Bayesian algorithm is "extremely theoretically tractable." While it is easy to state the optimization objective, its theoretical analysis doesn't seem to be tractable for general deep neural networks (which is not surprising as the maximal-update parametrization is also hard to theoretically analyze after some gradient-descent iterations). In fact, as explained in the paper itself, to solve for the maximum of the objective in Eq.(17)/Eq.(21), in general one must resort to numerical/variational methods.
>
> We have deleted this claim, and refocused the paper around the DGP results.  The DGP results really are surprisingly simple.  For instance, we show that the true DGP posterior is multivariate Gaussian (though of course this not true for DNNs).
>
> > Similarly (but in an opposite direction), the large body of work listed in the second paragraph of Section 2 -- which utilizes perturbation theory to analyze large-but-finite-width neural networks -- is dismissed as "highly complex." However, in these perturbative approaches, one can systematically understand the behavior of networks order by order in 1/width, and get layer-by-layer recursive equations -- much like that for the NNGP in the infinite-width limit -- that determine the depth dependences of these perturbative corrections. This is theoretically analyzable -- hence such a large body of work exists -- and dismissing them as "highly complex" seems misleading (especially because the paper uses such a claim to further illustrate the simplicity of the proposed representation learning limit).
>
> We have greatly limited these claims in the Related Work, and instead focus on our new DGP results.  (Any thoughts on how to state differences with the BNN setting would be appreciated).
>
> > (A-i) "elegant objective" in the abstract: it is not clear what makes Eqs.(17)/(21) "elegant."
>
> We have replaced this with an explicit description of the DGP objective as a combining a log-likelihood, and a series of KL-divergences between multivariate Gaussians.
>
> > (A-ii) "strong regulariser" in 3.4: it is not clear what makes it "strong" (in fact, the relative width ratio, \nu_{\ell}=N_{\ell}/N, in front of each KL-divergence term controls its strength, so the relative strength of the regularizer can be adjusted).
>
> We have deleted "strong".  We originally included it because the Gram matrices are arbitrary positive definite matrices, so the regulariser really does need to be "strong" to get them to behave sensibly.
>
> > (A-iii) "extremely theoretically tractable" in the abstract: "extremely" is subjective (even if we temporarily forget the previous major objection to this claim).
>
> Gone.
>
> > A missing period for "which we call the deep kernel machine (DKM) Here" in 3.7. (Also, the terminology/abbreviation is already introduced long before and used throughout the paper, so this phrasing doesn't make sense here.)
>
> Fixed.
>
> > C. In the first paragraph, for "(NNGP Matthews et al., 2018; Novak et al., 2018)" it is probably more standard to cite [Lee et al., 2017] as well (or in place of [Novak et al., 2018]). (I am not an author of these papers.)
>
> Fixed.
>
> D. It may be appropriate to also cite [Dyer and Gur-Ari, arXiv:1909.11304] and [Hanin and Nica, arXiv:1909.05989] in the second paragraph of Section 2. (I am not an author of these papers.)
>
> Fixed.
>
> > The proposed modification to the Bayesian algorithm is novel/original. However, the paper was not written in a clear manner. Here are a few suggestions (though far from exhaustive).
>
> We have greatly simplified the exposition by focusing the main text on the DGP results, and relegating the BNN extension to Appendix A.
>
> > I. On the core trick: the core trick of (noticing diminishing effects of target labels on posterior hidden-layer representations and then amplifying them by) replicating labels can be stated simply, yet it is not clear until a reader gets to the paragraph surrounding Eq.(16) buried in Sec.3.3. It would be beneficial/motivating to have it explained up front in Section 1.
>
> Fixed (we have a note about the key trick in the new contributions section).
>
> > II. On large datasets: since the paper makes a claim about scalability of the algorithm to large datasets, it would be beneficial to (i) have a discussion of how the computational time scales with the training dataset size and (ii) to detail how many training samples there are in each training dataset used in the paper.
>
> We have added a column with $P$, the number of datapoints, to table 1.  The sparse DKM scales linearly in the number of datapoints, as opposed to cubic scaling of the standard method.

---

### Official Review · Reviewer_JjRZ · 2022-10-31

**Confidence:** 3
**Clarity, Quality, Novelty And Reproducibility:** 1. Many strong claims from the paper …
**Correctness:** 2
**Technical Novelty And Significance:** 2
**Empirical Novelty And Significance:** 2
**Recommendation:** 5

**Strength And Weaknesses:**

# Strength.
1. The paper proposed simple and interesting ideas that allow feature learning in the infinite-width network, which is worth further exploration.
2. Some experiments from the paper show non-Gaussian behavior of the learned representation that differs from the NNGP limit.

# Weakness

1. The notations and terminologies are confusing, which makes it hard to parse the paper.
2. Missing key empirical comparison (vs. NNGP/NTK) for representation learning. I would like to see a comparison using CIFAR10.
3. Several strong claims that require further justifications.

**Summary Of The Paper:**

Large-width networks converge to GPs with fixed/ unlearnable kernels, making the networks unable to perform representation learning. In this paper, the authors proposed a simple recipe to allow for representation learning: making the logit layer as wide as the hidden layers and replicating the labels accordingly. The authors argue that this approach allows the kernel to evolve and possible learning a representation. The authors also support their claims using synthetical dataset experiments and UCI datasets.


**Summary Of The Review:**

Update:
I thank the authors for the clarification/updates of several notations/terminologies. I like the idea from the paper, a Bayesian prospective of feature learning. I have increased my score accordingly. However, there are still a lot of room for improvement and several open questions. To name some,

1. Even though this is a theory paper, authors should not shy away from more empirical work. Even doing a small scale experiments on Cifar10 (baseline against kernel) could really helpful to both practitioners and theorists.

2. Authors should have more detailed discussion/comparison between the meanfield/feature learning limit of neural networks and the current proposed Bayesian framework, both theoretically and empirically. E.g., baselining performance (bayesian feature learning vs NN feature learning), are they learning similar features, etc.


---------------------------------

Overall, I think the idea from the paper is interesting and worth further exploration. However, I also find the paper hard to parse due to clarity issues from notations and terminology, and the theoretical and empirical results from the paper do not support the strong claims from the paper.

---

> ### Author Response · Authors · 2022-11-10
> **Response (2/2)**
>
> > I would like to re-emphasize that the dataset used here (UCI) is too simple to capture representation learning. I would expect a comparison using CIFAR-10 against benchmark results for NNGP kernels; see Lee et al. finite vs infinite (https://arxiv.org/abs/2007.15801); neural kernel without tangents (Shankar, https://arxiv.org/abs/2003.02237). I am also interested in the visualization of the learned representation using the image dataset.
>
> Our work is not an empirical investigation, like Lee et al. (2020).  Instead, its contributions are almost entirely theoretical. We "capture representation learning" in the sense that we provide a simple objective (just a log-likelihood plus a series of KL divergences between multivariate Gaussians) that, when optimized, gives us the posterior representations from a DGP in the Bayesian representation learning limit.  Our experiments simply verify that this theory is correct.  For instance, Fig 2 (middle two rows) shows that the initial and trained DGP representations are different, and therefore that a DGP exhibits learned representations. Fig 2 (bottom two rows) shows that we "capture representation learning" in the sense that the trained DKM exactly matches the learned representation in the DGP (bottom two rows).
>
> > In addition, Eq (13) seems to be a result in "Wide Bayesian neural networks have a simple weight posterior: theory and accelerated sampling" by Hron et al. Please clarify.
>
> Of course it is!  At the start of that section, we say "We are now in a position to take a new viewpoint on the DGP analogue of standard NNGP results (Lee et al., 2017; Matthews et al., 2018)."  And we have added Hron et al. to that list.
>
> Notations and Terminologies
>
> > The definition of DNNs is confusing in Sec 3. Do you mean a fully-connected network with trainable parameters that are optimized using gradient descent? This is what DNNs mean for most people in ML.
>
> The main results are for DGPs, so we have pushed all the neural network results to Appendix A.  Additionally, for clarity, we have changed DNN -> Bayesian (BNN).  i.e. its a fully connected network with weights that are learned using Bayesian inference.  To
>
> > In addition, what does that mean to marginalize the weights in each layer in eq (2)? Do you mean sequentially?
>
> Integrating out the prior on the weights is described in depth in Appendix A.  In short, the standard prior specifies $P(W_\ell)$ and $P(F_\ell| W_\ell, F_{\ell-1})$.  To get 1a, we need to combine these two, and integrate over the weights,
>
> $P(F_\ell| F_{\ell-1}) = \int dW_\ell P(F_\ell| W_\ell, F_{\ell-1}) P(W_\ell)$
>
> We're not quite sure what sequentially refers to here?
>
> > Can you verbally explain the difference between (3a) and (3b)?
>
> Eq. 3b (now Eq. 3) is just the average outer product of the $f_\lambda^{\ell-1}$s.  3a (now Eq. 30) is the same thing, but where we first apply the neural network nonlinearity, $\phi$ (e.g. relu).
>
> > DGP "Deep Gaussian Processes" (by Andreas C. Damianou, Neil D. Lawrence, not cited in the paper) is a standard framework. Is it the same thing used in the paper? If so, why not cite the above (or related ) paper?
>
> We have added this citation.  (We only cited Salimbeni and Diesenroth originally, because our derivations were initially based on the doubly stochastic framework).
>
> > Notations in equation (11) are confusing. What are $G_l$/$G_{l−1}$ here? Don't they depend on $N$? Or are they some infinite/deterministic objects that don't rely on $N$, and you overload the notations? Same for equations (12), etc.
>
> When considered as a random variable, $G_\ell$ does depend on $N_\ell$ (see Eq. 7).  In contrast, in Eq. (11), we're evaluating the posterior probability at a particular value of $G_1,...,G_L$ that we get to choose.  We would usually fix this confusion by using e.g. $X$ for the random variable, and $x$ for the setting of the random variable that we evaluate the probability at.  For instance,
> \begin{align}
>   X \sim \mathcal{N}(\mu, 1)
> \end{align}
> Of course, the random variable $X$ depends on $\mu$.  But if we were to compute the log probability,
> \begin{align}
>   \log P(X=x) &= - \tfrac{1}{2} \log 2 \pi - \tfrac{1}{2} (x - \mu)^2,
> \end{align}
> we would get to choose the value of $x$, and the value of $x$ that we choose to evaluate at of course does not depend on $\mu$.  Unfortunately, it did not make sense to use this notation in our context, because the G's are matrices, so they should really always be capitalised.
>
> > First citation in the intro. Please cite "Radford M. Neal. Priors for infinite networks," which is the first NNGP (one-hidden layer) paper. "Lee et al Deep Neural Networks as Gaussian Processes" should also be added.
>
> Fixed.

---

> ### Author Response · Authors · 2022-11-10
> **Response (1/2)**
>
> > I can't agree with the claim that the approach is extremely theoretically tractable. (in the abstract) and it is misleading. Unlike NNGP, the solution can be written analytically as an exact closed-form formula of the input data. The approach in the paper is not analytically tractable (except possibly in the linear network setting) and requires approximation methods. Cited from the paper, "In practice, the true posteriors required to evaluate Eq. (21) are intractable ".
>
> We cannot possibly hope for anything closed form like the NNGP, as representation learning is a _much_ harder problem. Nonetheless, for DGPs, we have a solution that is exact, and at the same time much simpler than all the other present in the literature. In particular, we show that the true DGP posteriors are multivariate Gaussian (Eq. 19; Fig 3). And the exact Gaussian covariances can be found by optimizing an objective that combines a log-likelihood term, and a series of KL-divergences between multivariate Gaussians. Of course, we do additionally give the NN results, which are more complicated (and for that reason, we have relegated these results to an Appendix).  We have edited the abstract to emphasise these points (but note these changes only appear in the pdf).
>
> > Using **the** representation learning limit is not proper; there can be many of them, and the proposed approach is just one of many.
>
> Of course there are other such limits for the NTK, such as "the feature learning limit". But it does seem very unlikely that other natural limits exist in the Bayesian setting, as Bayes theorem tightly restricts what we're doing. Certainly, if we only allow ourselves to scale the prior or likelihood log-probability, this would seem to be the only natural limit.  As such, we've switched to the "Bayesian representation learning limit".
>
> > "We show that DKMs can be scaled to large datasets using inducing point methods from the Gaussian process literature, and we show that DKMs exhibit superior performance to other kernel-based approaches" (from the abstract.) This is certainly an overclaim. The largest dataset used in UCI, could not justify superior performance and scalability to large datasets. Again, I expect at least experimental results on CIFAR10 if not more complicated.
>
> We have updated these sentences to: "Next, we introduce the possibility of using this limit and objective as a flexible, deep generalisation of kernel methods, that we call deep kernel machines (DKMs). Like most naive kernel methods, compute for DKMs scales cubically in the number of datapoints. We therefore develop a sparse DKM using methods from the inducing point literature that scales linearly in the number of datapoints, opening the possibility of practically applying these systems."  (Note these changes appear only in the pdf).  This focuses in on the point of the exercise, which is to introduce an entirely new class of deep kernel method, not to argue about performance on particular datasets.

---

### Decision · Program_Chairs · 2023-01-20

**Decision:**

Reject

**Justification For Why Not Higher Score:**

Authors did address some weaknesses in the rebuttal and reviewers acknowledge there is an interesting idea for further exploration. But the paper needs to improve clarity and readability and should put the result with the existing mean field and maximal update limit. Given the expert reviewers having hard time clearly seeing the significance means presentation needs to be much improved for a broad ICLR audience. While AC believes this paper has good potential and interesting ideas, given current evaluation of reviewers, the paper is not ready for acceptance.

**Justification For Why Not Lower Score:**

N/A

**Metareview: Summary, Strengths And Weaknesses:**

Conventional infinite-width limits of neural networks become NNGP and this limit does not have feature learning. The author proposes a simple recipe for a fixed-depth-infinite-width limit neural network  to perform Bayesian representation learning. This is done by making the logit layer to be as wide as the hidden layers by replicating labels.

Strength
- Proposes simple and non-trivial interesting idea allowing feature learning in infinite-width network, which is worth of further exploration
- Experiments show non-Gaussian behavior of learned representation which is indication that representation differs from NNGP
- Provides Bayesian perspective of feature learning in wide neural networks which is important topic

Weakness
- Clarity in terms of notation, terminology, subjective adjective/adverbs: resulting in poor readability
- Missing key empirical comparison to NNGP / NTK and in natural data like CIFAR-10.
- Missing discussion / comparison to other feature learning limits of neural networks.  (e.g. mean field limits and maximal update parameterization)
- Theoretical and empirical results in the paper do not support the strong claims.